# Temperature-driven vapor pressure deficit structures forest

# 2 bryophyte communities across the landscape

- 3 Anna Růžičková <sup>1,2</sup>, Matěj Man <sup>1,2</sup>, Martin Macek <sup>1</sup>, Jan Wild <sup>1</sup>, Martin Kopecký <sup>1</sup>
- <sup>1</sup> Institute of Botany of the Czech Academy of Sciences, Zámek 1, Průhonice, CZ-252 43, Czech Republic,
- <sup>2</sup> Department of Botany, Faculty of Science, Charles University, Benátská 2, Prague 2, CZ-128 00, Czech Republic
- 6 Correspondence to: Anna Růžičková (anna.ruzickova @ibot.cas.cz), Martin Kopecký (ma.kopecky@gmail.com)

### Abstract

- Atmospheric vapor pressure deficit (VPD) controls local plant physiology and global vegetation productivity.
- However, at ecologically crucial intermediate spatial scales, the role of VPD variability in forest bryophyte
- community assembly and the processes controlling this variability are little known.
- To explore VPD effects on bryophyte community composition and richness and to disentangle processes
- controlling landscape-scale VPD variability, we recorded bryophyte communities and simultaneously measured
- forest microclimate air temperature and relative humidity across a topographically diverse landscape representing
- a bryophyte diversity hotspot in temperate Europe. Based on VPD importance for plant physiology, we
- hypothesize that VPD can be important also for bryophyte community assembly and that VPD variability will be
- jointly driven by saturated and actual vapor pressure across the topographically diverse landscape with contrasting
- forest types and steep microclimatic gradients.
- Contrary to our expectation, VPD variability was dictated by temperature-driven differences in saturated vapor
- pressure, while actual vapor pressure was surprisingly constant across the landscape. Gradients in species
- composition, species richness and community structure of bryophyte assemblages followed closely the VPD
- variability. The average daily mean VPD was much better predictor of species composition than average daily
- maximum VPD and the mean VPD also explained significantly more variation in species composition and richness
- than maximum temperature, indicating that time-averaged evaporative stress is more relevant for bryophyte
- communities than microclimatic extremes. While mesic forest bryophytes occurred along the whole VPD gradient,
- species occurring near their distributional limits and locally rare species preferred sites with low VPD. In result,
- low VPD sites represent species-rich microclimatic refugia within the landscape, where regionally abundant mesic
- forest bryophytes coexist with rare species occurring near their distributional range limits.
- Our results showed that VPD variability at ecologically crucial landscape scales is controlled
- by temperature-driven saturated vapor pressure. Future climate warming will thus increase evaporative stress and
- reshuffle VPD-sensitive forest bryophyte communities even in topographically diverse landscapes, which are
- traditionally considered as microclimatic refugia buffered against climate change. Bryophyte species occurring
- near their distributional range limits concentrated in low VPD sites will be especially vulnerable to the future
- changes in atmospheric VPD.

### 1. Introduction

73

Vapor pressure deficit (VPD) expresses atmospheric water demand as the difference between the amount of water 36 vapor the air can hold at a given temperature and the actual amount of water vapor present in the air. Unlike relative 37 air humidity, VPD accurately expresses plant evaporative stress (Campbell & Norman 1997). Since air capacity to hold water vapor increases exponentially with temperature, the same relative humidity at different temperatures 38 39 indicates very different atmospheric moisture conditions (Anderson, 1936). An atmosphere with the same relative 40 air humidity may be very "dry" (when the temperature is high) or it may be very "wet" (when the temperature is 41 low). Relative air humidity therefore does not indicate the atmospheric moisture condition in physiologically 42 meaningful way, despite its popularity in ecological studies (Campbell & Norman 1997). In contrast, VPD directly 43 expresses the atmospheric moisture conditions in terms of plant evaporative stress (Anderson, 1936). 44 Atmospheric VPD is a key driver of plant functioning in terrestrial ecosystems (Ruehr et al., 2014; Grossiord et 45 al., 2020), because higher VPD leads to reduced photosynthesis in the short term and drought-induced mortality 46 in the long term (McDowell et al., 2008; Fu et al., 2022). Ongoing climate changes exacerbate VPD-driven 47 evaporative stress because higher temperatures lead to an exponential increase in VPD (Lawrence, 2005; Grossiord 48 et al., 2020). Increasing VPD already limits global vegetation productivity (Yuan et al., 2019; López et al., 2021; 49 Lu et al., 2022) and triggers large-scale forest diebacks (Breshears et al., 2013; Eamus et al., 2013; Williams et al., 50 2013). Yet, in contrast to the widely recognized role of VPD in local plant physiology and global vegetation 51 functioning, VPD effects on plant community assembly are largely unknown (Novick et al., 2024). 52 The knowledge about VPD effects on plant communities and the processes that control VPD variability over 53 the landscape are crucial for more realistic predictions of climate change impacts on vegetation 54 and the identification of microclimatic refugia (Ashcroft and Gollan, 2013; Davis et al., 2019; Finocchiaro et al., 55 2024; Ogée et al., 2024). Because VPD is a difference between saturated vapor pressure (Psat) and actual vapor pressure (Pair), VPD variability reflects the interplay between spatial patterns in saturated and actual vapor 56 57 pressures. While saturated vapor pressure is solely an exponential function of air temperature, actual vapor 58 pressure is influenced by many processes operating at different spatial scales - ranging from regional atmospheric 59 circulation and precipitation to local evaporation and plant transpiration (Campbell and Norman, 1998). Yet, 60 despite increasingly recognized VPD importance, it is still unknown how these contrasting processes integrate into 61 the VPD variability over the landscape. 62 A deeper understanding of the mechanisms behind landscape-scale VPD variability is particularly important for 63 climate change biology. Scientists predict a temperature increase of up to 4.4 °C by 2100 (IPCC, 2023), which 64 would lead to a more than 40 % increase in VPD for the same atmospheric water vapor content (Will et al., 2013). 65 These changes can also modify VPD variability over the landscape, potentially shift the distribution of individual 66 species and therefore alter the composition of plant communities. However, VPD effects on plant distribution 67 and community assembly over the landscape are not sufficiently known. 68 Among plants, bryophytes are exceptionally sensitive to evaporative stress because they lack roots, lignified 69 water-conducting system, water storage tissues, and active stomata and have a large surface area in proportion to 70 biomass (Rice et al., 2001, Goffinet and Shaw, 2009). When exposed to the air with non-zero VPD, bryophytes 71 therefore inevitably lose water (Hinshiri and Proctor, 1971; Busby and Whitfield, 1978). Because bryophytes 72 transport water only passively, mainly through external capillary spaces between tiny parts of their body

(Schofield, 1981), their internal water content is a function of the water availability in the surrounding

environment (Vanderpoorten and Goffinet, 2009). Once the external water evaporates, bryophyte cells rapidly lose turgor, metabolic activity slows down, and carbon fixation decreases.

To cope with this evaporative stress, bryophytes developed evolutionary and ecologically unique desiccation strategy, allowing them to survive drought episodes in a desiccated state (Proctor, 2000, 2001). Despite this ability to survive microclimatic extremes, bryophyte assemblages are potentially sensitive to evaporative stress, because desiccation tolerance widely differs among bryophyte species (Hinshiri and Proctor, 1971; Wagner and Titus, 1984, Oliver et al., 2000; Proctor, Ligrone, et al., 2007; Proctor, Oliver, et al., 2007). Therefore, it can be assumed that the atmospheric VPD – an ecologically meaningful variable expressing evaporative stress – will strongly affect composition, richness and structure of bryophyte assemblages. Yet surprisingly little is known about the VPD effect on bryophyte assemblages in temperate forests (Fenton and Frego, 2005).

To provide this missing knowledge, we combined detailed in-situ forest microclimate measurements with simultaneous bryophyte inventories conducted across topographically diverse landscape representing bryophyte diversity hotspot in central Europe. Using these data, we explored how landscape-scale VPD variability affects

diversity hotspot in central Europe. Using these data, we explored how landscape-scale VPD variability affects bryophyte community composition and species richness in temperate forests, quantified VPD variability over the topographically diverse landscape, and identified which processes drive this variability.

### 2. Material and methods

## 2.1 Study area

We recorded bryophytes and measured microclimate in the Bohemian Switzerland National Park in the Czech Republic (Fig. 1). The rugged terrain of this sandstone landscape creates a fine-scale mosaic of contrasting habitats with steep microclimatic gradients over short distances (Wild et al., 2013). The elevation within the national park ranges from 125 to 619 m, and the mean elevation is 340 m. According to the data from the Tokáň weather station (Fig. 1), the mean annual air temperature during the 2011-2019 period was 8.3 °C, and the mean annual precipitation was 765 mm.

Figure 1: We measured microclimate and simultaneously recorded bryophyte species composition at 38 permanent research plots within the Bohemian Switzerland National Park in Central Europe. This forested area has rugged terrain creating steep environmental gradients over short distances.

- Most of the Bohemian Switzerland is covered with coniferous forests. Norway spruce (*Picea abies*) planted mostly
- during the 19th and 20th century dominates in the valleys and on the plateaus, while patches of semi-natural forests
- are dominated either by Scots pine (*Pinus sylvestris*) on the upper slopes and rocky ridges or by European beech
- (Fagus sylvatica) on more mesic sites.
- The nutrient-poor and strongly acidic soils result in a relatively low diversity of vascular plants, which contrasts
- with the exceptionally rich bryophyte flora (Härtel et al., 2007). The Bohemian Switzerland currently hosts more
- than 300 bryophyte species, and therefore represents a hotspot of bryophyte diversity in Central Europe (Marková,
- 2008).

113

131

- The bryophyte flora of the Bohemian Switzerland is dominated by forest species like Tetraphis pellucida,
- Bazzania trilobata, and Dicranum scoparium. These dominant floristic elements are enriched by disjunct
- occurrences of (sub)alpine or (sub)montane (e.g., Hygrobiella laxifolia, Geocalyx graveolens, Anastrophyllum
- michauxii), boreal (e.g., Dicranum majus, Rhytidiadelphus subpinnatus), and (sub)oceanic (e.g., Tetrodontium
- brownianum, Plagiothecium undulatum) species (Härtel et al., 2007; Marková, 2008).

### 2.2 Field data collection

- We recorded bryophyte species composition and measured microclimate on 38 permanent plots within
- the Bohemian Switzerland National Park (Fig. 1). These plots were selected through stratified-random sampling
- to capture the main microclimatic gradients within the core zone of the national park. Specifically, using GIS
- and LiDAR-based digital terrain model, we first divided the study area into geographical strata defined by
- the terrain (valley bottoms, lower slopes, upper slopes, and ridges) and further separated the slopes with
- predominantly northern and southern orientation. Within each stratum, we randomly selected an equal number of
- locations separated by at least 50 m. In the field, we navigated to the selected location with GPS device and placed
- the center of plot 1.5 m to the north from the nearest tree.
- Within each permanent plot, we installed HOBO U23 ProV2 (Onset, USA) microclimatic datalogger at 1.5 m
- height on the north side of a tree nearest to the plot center. Each HOBO datalogger was protected by a white
- radiation shield with good ventilation and measured air temperature (resolution 0.02 °C, accuracy  $\pm 0.21$  °C)
- and relative humidity (resolution 0.05 %, accuracy  $\pm$  2.5 %) every 30 minutes from 1 June to 31 August 2022.
- Simultaneously with microclimate measurements, we recorded the presence of all bryophyte species in each
- research plot following the nomenclature of the Czech national checklist (Kučera et al., 2012). We deliberately
- sampled bryophytes in a relatively small circular plot with 1 m radius (3.14 m²) without any exposed rocks or big
- stones to reduce the possible effects of within-plot environmental heterogeneity (Rambo and Muir, 1998;
- Vanderpoorten and Engels, 2002; Schmalholz and Hylander, 2011).

### 2.3 Microclimate data processing

- First, we checked the microclimatic time series visually and then with standard automated procedures implemented
- in the *myClim* R package (Man et al., 2023). Air humidity measurement with microclimatic loggers is sensitive to
- water condensation, resulting in unrealistically high measurements for prolong periods of time (Ashcroft and
- Gollan, 2013; Feld et al., 2013). We therefore carefully checked microclimatic time series and found no signs of
- the condensation effect.

Using checked air temperature and relative humidity data, we first calculated the saturated vapor pressure (P<sub>sat</sub>)

following the updated Buck formula (Buck, 1981, 1996):

 $139 \qquad P_{sat} = \; (1.003 + 4.18 \times 10^{-6} \times 101 \; kPa) \times 0.61115 \times e^{((23.036 - t/333.7)*(t/(279.82 + t)))},$ 

where *t* is air temperature [ $^{\circ}$ C].

Then, we calculated the actual vapor pressure (Pair) using the Tetens's formula (Tetens, 1930):

$$P_{air} = P_{sat} \times \left(\frac{rh}{100}\right)$$
,

where rh is relative humidity [%].

Finally, we calculated atmospheric VPD as the difference between P<sub>sat</sub> and P<sub>air</sub> (Jones, 2014). Using the resulting microclimatic time series, we calculated three variables representing evaporative stress (Tab. 1). First, we calculated the average daily maximum temperature (T<sub>max</sub>). While T<sub>max</sub> is ecologically less meaningful proxy for evaporative stress than atmospheric VPD (Campbell and Norman, 1998; Eamus et al., 2013), several previous studies identified T<sub>max</sub> as highly relevant microclimatic variable linked to evaporative stress and affecting species composition and richness of forest vascular plants and bryophytes within the central Europe (Macek et al., 2019, Man et al., 2022). Then, we calculated two variables capturing different aspects of VPD driven evaporative stress. First, we calculated the average daily maximum VPD (VPD<sub>max</sub>), which represents site-specific microclimatic extremes (Ashcroft and Gollan, 2013). Second, we calculated the average daily mean VPD, which represents time-aggregated evaporative demand experienced by bryophytes on each site.

To disentangle the drivers of spatio-temporal VPD variability over the landscape, we calculated also plot-specific daily average values of  $P_{\text{sat}}$  and  $P_{\text{air}}$  (Tab. 1).

Table 1: Overview of microclimatic variables representing evaporative stress (T<sub>max</sub>, VPD<sub>max</sub>, VPD<sub>mean</sub>) and its components (P<sub>sat</sub>, P<sub>air</sub>). For each variable, we provide the overall mean and range of plot-specific averaged daily values measured continually during summer 2022 on 38 forest research plots in the Bohemian Switzerland National Park, Czech Republic.

| Microclimatic variable         | Abbreviation                  | Overall mean | Range of plot means |
|--------------------------------|-------------------------------|--------------|---------------------|
| Maximum air temperature        | $T_{max}$                     | 24.26 °C     | 18.80–27.64°C       |
| Maximum vapor pressure deficit | $\mathrm{VPD}_{\mathrm{max}}$ | 2.09 kPa     | 0.62–3.17 kPa       |
| Mean vapor pressure deficit    | $VPD_{mean}$                  | 0.85 kPa     | 0.23–1.16 kPa       |
| Mean saturated vapor pressure  | $\mathbf{P}_{\mathrm{sat}}$   | 2.63 kPa     | 2.09–2.93 kPa       |
| Mean actual vapor pressure     | $P_{air}$                     | 1.78 kPa     | 1.66–1.90 kPa       |

## 2.4 Data analysis

### 2.4.1 Bryophyte community composition, richness and structure

In our analysis, we focused on the relationship between microclimatic variables representing evaporative stress and bryophyte community composition, structure, and richness. First, we identified the main gradients in community composition and explored their relationship with variables representing evaporative stress. Then, to explore which variable representing evaporative stress is more closely associated with bryophyte community composition and richness, we calculated the variability in species composition and richness explained by the mean and maximum atmospheric VPD and maximum air temperature. Further, to disentangle the effects of atmospheric

VPD from the effects of the maximum temperature, we partition the explained variability into independent 169 and shared fractions. Finally, we tested the link between VPD and bryophyte community structure through 170 nestedness analysis. To explore the main gradients in the bryophyte community composition, we used non-metric multidimensional 171 172 scaling (NMDS) to extract the main patterns in bryophyte community composition expressed with the Sørensen 173 dissimilarity index. We calculated two-dimensional NMDS with the weak treatment of ties, a maximum of 500 174 random starts, and 999 iterations in each NMDS run using metaMDS function from the vegan R package version 175 2.6-4 (Oksanen et al., 2022). To maximize variance along the first ordination axis, we centered and rotated 176 the resulting two-dimensional configuration with principal component analysis. 177 To explore how main compositional gradients correlate with microclimate variables representing evaporative 178 stress, we passively projected vectors of maximum and mean VPD, and maximum temperature into the NMDS 179 ordination space and tested the significance of the fit with 999 random permutations using the *envfit* function from vegan R package (Oksanen et al., 2022). Finally, we projected bryophyte species richness gradients into the NMDS 180 181 ordination space using a generalized additive model fitted through ordisurf function from vegan R package 182 (Oksanen et al., 2022). To quantify the relationship between the microclimatic variables representing evaporative 183 stress and species richness expressed as a number of bryophyte species recorded in the plot, we used a generalized 184 additive model (GAM) fitted with the R package mgcv 1.9.1 (Wood, 2011). We used GAM with Poisson 185 distribution, log link function, and smooth terms fitted by thin plate regression splines without null space 186 penalization and smoothing parameter estimation using restricted maximum likelihood. To assess the statistical 187 significance, we used a  $\chi^2$  test comparing the fitted model to the only intercept null model. 188 To calculate the proportion of variability in bryophyte community composition explained by microclimatic 189 variables representing evaporative stress, we used distance-based RDA (McArdle and Anderson, 2001) calculated 190 on the same Sørensen dissimilarity matrix as used for NMDS. We calculated the distance-based RDA (db-RDA) 191 with dbrda function from vegan R package (Oksanen et al., 2022) and assess the statistical significance using 999 192 random permutations of the raw data (Legendre et al., 2011). 193 As all three microclimatic variables representing evaporative stress were correlated (Appendix A), we explored 194 their shared and independent effects on bryophyte community composition and richness through variation 195 partitioning (Legendre, 2008). Because VPD<sub>max</sub> and T<sub>max</sub> were almost identical (Pearson R = 0.98), we disentangled shared and independent effects of substantially less correlated VPD<sub>mean</sub> and T<sub>max</sub> 196 197 (Pearson R = 0.78). To quantify their independent and shared effects, we partitioned the variation in bryophyte 198 community composition explained by atmospheric VPD<sub>mean</sub> and T<sub>max</sub> using adjusted R<sup>2</sup> (Peres-Neto et al., 2006) 199 calculated with the *varpart* function from the *vegan* R package (Oksanen et al., 2022). 200 To quantify the shared and independent effects of atmospheric VPD<sub>mean</sub> and T<sub>max</sub> on species richness, 201 we partitioned the deviance explained in GAM models. First, we related species richness to atmospheric VPD<sub>mean</sub> 202 and T<sub>max</sub> in the full GAM, when both variables were used simultaneously as predictors. Then, we fitted two partial 203 GAMs (first with VPD<sub>mean</sub>, second with T<sub>max</sub> as explanatory variables). To prevent different smoothing parameters 204 in the partial models, we extracted smoothing parameters from the full GAM and used them in both partial GAMs 205 (Hjort et al., 2012). To assess the statistical significance, we compared each model against the null model with

6

only intercept using a  $\chi^2$  test. To assesses the significance of the independent effects of atmospheric VPD and  $T_{max}$ .

we compared partial GAMs with the full GAM using  $\chi^2$  test.

- Finally, we used nestedness analyses (Ulrich et al., 2009) to test the VPD effects on bryophyte community
- structure. To directly test the two hypotheses about the bryophyte community structure along the VPD gradient,
- we first order the community matrix along the gradient of increasing plot-specific VPD<sub>mean.</sub> To test the first
- hypothesis that the bryophyte communities from sites with high VPD are nested subsets of bryophyte communities
- from sites with low VPD, we used NODF<sub>sites</sub> metric (Almeida-Neto et al., 2008). To test the second hypothesis
- that more frequent bryophyte species occur along the whole VPD gradient, but less frequent species are
- concentrated on sites with low VPD, we used NODF<sub>species</sub> metric (Almeida-Neto et al., 2008). To calculate both
- NODF metrics, we used *nestednodf* function from the vegan R package (Oksanen et al., 2022).
- We used a null model approach to assess the statistical significance of nestedness patterns (Ulrich et al., 2009).
- Specifically, we compared the observed NODF values to the distribution of 999 NODF values calculated through
- the conservative R1 null model, which maintains species richness of the site and uses species frequencies as
- probabilities of selecting species (Wright et al., 1997). To quantify the difference between the observed NODF
- values and the NODF values generated by the R1 null model, we calculated the standardized effect size (SES)
- expressing the number of standard deviations that the observed NODF value differs from the mean NODF value
- of the simulated assemblages (Ulrich et al., 2009). To construct the null models and to calculate SES, we used
- the *oecosimu* function from the vegan R package (Oksanen et al., 2022).
- We used R version 4.4.0 (R Core Team, 2024) for complete data analysis and figure preparation. For
- the color-blind safe gradients we used the R package scico 1.5.0 (Pedersen and Crameri, 2023).

# 226 2.4.2 VPD variability across the landscape

- Using the time-series of both VPD components (P<sub>sat</sub> and P<sub>air</sub>), we explored their spatio-temporal variability
- and quantify their influence on the VPD variability over the landscape. First, we explored how variable was VPD
- and both its components over the landscape in a daily timesteps. Then, we averaged this daily variability into
- the overall measure of spatial variability in VPD, P<sub>sat</sub>, and P<sub>air</sub> during the whole study period. Finally, we used
- variation partitioning to quantify how much was VPD variability controlled by P<sub>sat</sub> and P<sub>air</sub>.
- To quantify spatial variability in daily VPD and both its components (P<sub>sat</sub> and P<sub>air</sub>) over the landscape, we calculated
- the standard deviation (SD) of the plot-specific daily mean VPD, P<sub>sat</sub> and P<sub>air</sub> values among all study plots. In this
- first step, we calculated SD of these microclimatic variables for every day within the study period separately. Then
- we averaged these daily inter-plot SD values separately for VPD, P<sub>sat</sub> and P<sub>air</sub> into an overall measure of spatial
- variability for each microclimatic variable during the whole study period.
- Finally, to disentangle the contribution of P<sub>sat</sub> and P<sub>air</sub> to the VPD variability over the landscape, we performed
- variation partitioning based on a multiple linear regression model and adjusted R2 (Legendre, 2008) with
- the plot-specific mean VPD as the response variable and the mean  $P_{sat}$  and  $P_{air}$  as the predictors.

### 3. Results

## 241 3.1 Bryophyte community composition, richness and structure

- In total, we recorded 39 bryophyte species: 14 liverworts and 25 mosses (Appendix C, Tab. C1). The species
- richness was highly variable among the plots while the average number of species per plot was 8, the minimum

was 1 and the maximum 21. The most frequent species were Dicranum scoparium (n = 32), Leucobryum juniperoideum (n = 26) and Hypnum cupressiforme (n = 24). Main patterns in community composition and species richness reflected the gradient of evaporative stress (Fig. 2). Gradient in T<sub>max</sub> was highly correlated to the gradients in VPD (Fig. 2), but main patterns in community composition were less related to  $T_{max}$  than to VPD (vegan::envfit -  $T_{max}$ :  $R^2 = 0.32$ , p = 0.003; *vegan*::*envfit* – VPD<sub>mean</sub>:  $R^2 = 0.52$ , p = 0.001; VPD<sub>max</sub>:  $R^2 = 0.37$ , p = 0.001). The number of bryophyte species was higher in plots with low VPD and declined with an increasing VPD (Fig. 2). Both atmospheric VPD and maximum temperature were significantly associated with species richness, but maximum temperature explained substantially less deviance (Table 2). The mean VPD explained slightly more deviance than the maximum VPD (Table 2).

Figure 2: Nonmetric multidimensional scaling (NMDS) of the bryophyte community composition showing main gradients in bryophyte assemblages sampled at 38 temperate forest plots. Points show the positions of the individual plots within the NMDS ordination space, and the vectors show the gradients in the maximum air temperature ( $T_{max}$ ), maximum VPD (VPD<sub>max</sub>) and mean VPD (VPD<sub>mean</sub>). The smooth surface and associated contours fitted into the NMDS ordination space with a generalized additive model show the pattern of decreasing species richness with increasing evaporative stress.

The mean VPD explained substantially more variation in species composition than the maximum VPD and the maximum temperature (Table 2). When used independently, both VPD<sub>mean</sub> and  $T_{max}$  were significant predictors of bryophyte community composition (Table 2). However, the effect of  $T_{max}$  almost completely overlaps with VPD<sub>mean</sub> (Fig. 3). When we controlled for the effect of mean VPD, maximum temperature did not explain significant part of variation in community composition (*vegan*::*dbrda* – adj.  $R^2 = 0$  %, p = 0.764) or in species richness (*mgcv*::*gam* –  $D^2 = 3.72$  %, p = 0.174). In contrast, the mean VPD explained a significant part of variation in species composition and richness even after the controlling for maximum temperature (*vegan*::*dbrda* – adj.  $R^2 = 6$  %, p = 0.003), see Fig. 3. Therefore, the mean VPD explained substantially more variation in bryophyte community composition and richness than maximum temperature and maximum temperature did not have any significant effects independent from the mean atmospheric VPD (Fig. 3).

Table 2: Variation in community composition and species richness explained by three microclimatic variables representing evaporative stress. To quantify variation explained by each variable, we used distance-based RDA (db-RDA) for community composition and generalized additive models (GAM) for species richness.

|                        | Community composition (db-RDA) |          | Species richness (GAM) |                            |       |         |
|------------------------|--------------------------------|----------|------------------------|----------------------------|-------|---------|
|                        | Variation (R <sup>2</sup> )    | pseudo-F | p-value                | Deviance (D <sup>2</sup> ) | χ2    | p-value |
| Microclimatic variable |                                |          |                        |                            |       |         |
| mean VPD               | 16.09 %                        | 6.90     | 0.001                  | 32.8 %                     | 27.04 | 

# species richness

Figure 3: Variation partitioning showing independent and shared effect of mean VPD (VPD<sub>mean</sub>) and maximum air temperature ( $T_{max}$ ) on bryophytes species composition and richness in 38 forest plots. Values represent adjusted  $R^2$  from db-RDA for species composition and explained deviance from GAM for species richness. While VPD<sub>mean</sub> has significant effects even after the controlling for  $T_{max}$  both for species composition (p = 0.003) and richness (p = < 0.001), the unique effects  $T_{max}$  was non-significant both for species composition (p = 0.764) and richness (p = 0.174).

Bryophyte community structure was closely related to the gradient of mean atmospheric VPD (Fig. 4). Bryophyte communities from plots with higher VPD were generally impoverished and compositionally nested subset of

281 the communities from sites with lower VPD (vegan::oecosimu - NODF<sub>sites</sub> =39.17, SES =4.26, p = 0.001). 282 Moreover, while frequent species occurred along the whole VPD gradient, rare species occurred preferably on sites with low VPD ( $vegan::oecosimu - NODF_{species} = 29.97$ , SES = 3.34, p =0.003). 283 At the species level, small liverworts (e.g. Riccardia multifida, Lophozia ventricosa) and hygrophilous bryophytes 284 285 (e.g. Polytrichum commune, Bazzania trilobata), as well as species with boreal (e.g. Dicranum majus) and (sub)oceanic (e.g. Mylia taylorii, Plagiothecium undulatum) distribution preferred plots with low atmospheric 286 287 VPD (Fig. 4). In contrast, regionally frequent species like Hypnum cupressiforme, Polytrichum formosum or Dicranum scoparium occurred also in plots with higher atmospheric VPD (Fig. 4). 288

Figure 4: Occurrences of all recorded bryophyte species along the gradient of the mean VPD measured at 38 forest plots. Plots are sorted from the lowest to highest mean VPD and each filled square shows the presence of the focal species within the plot. While rare and species near their distributional range limits prefer sites with low VPD, mesic forest species occur along the whole VPD gradient.

### 3.2 VPD variability across the landscape

VPD in the forest understory was highly variable across the landscape (Fig. 5). While the variability in saturated vapor pressure was comparable to the variability in VPD, actual vapor pressure was much less variable among

# Spatial variation in

Figure 5: Spatio-temporal variability of VPD and its components – saturated and actual atmospheric vapor pressures. Each data point shows the standard deviation of the plot-specific daily mean values simultaneously measured at 38 forest plots, and density plots summarize this spatio-temporal variability over the summer season. The individual data points were slightly jittered for better visibility.

The dominant driver of VPD variability across the landscape was temperature-driven saturated vapor pressure (Fig. 6). In a univariate linear regression model,  $P_{sat}$  explained 93 % of VPD variability, while  $P_{air}$  explained 30 %. However,  $P_{sat}$  and  $P_{air}$  were negatively correlated (Pearson R = -0.31) and variation partitioning based on multiple regression model showed that the  $P_{air}$  uniquely explained only 7 % of variability in VPD (Fig. 6). Therefore, temperature-driven  $P_{sat}$  was the dominant driver of VPD variability, while spatial variation in  $P_{air}$  contributed surprisingly little to the overall VPD variability across the landscape.

Figure 6: Atmospheric vapor pressure deficit (VPD) was driven by temperature-dependent saturated vapor pressure, while actual vapor pressure was weakly related to local VPD. Each dot represents the mean VPD, and the mean saturated and actual vapor pressure measured during the summer at 38 forest plots established over topographically diverse landscape. Venn diagram shows variation (adjusted  $R^2$ ) in mean VPD explained solely by mean saturated ( $P_{sat}$ ) and mean actual ( $P_{air}$ ) vapor pressure and the variation explained jointly by both predictors.

### 4. Discussion

We found that community composition and richness of forest bryophytes was significantly affected by atmospheric VPD. Our findings have important implications both for theoretical and applied ecology. First, the variation in VPD over the landscape was largely controlled by air temperature. Therefore, air temperature and VPD are tightly coupled at biologically relevant scales, and their effects are hard to disentangle with observational data. Interestingly, this coupling was strongest between maximum VPD and maximum temperature and maximum temperatures was previously identified as a key driver of bryophyte and vascular plant species distribution in temperate forests (Macek et al., 2019; Man et al., 2022). Unfortunately, these studies did not measure VPD. Considering our results, the importance of maximum temperature does not necessarily stem from its direct effects on plant ecophysiology, but more likely from strong temperature control of VPD variability over the landscape. Nevertheless, this new hypothesis needs further testing.

Interestingly, we also found that mean VPD was a much better predictor of bryophyte community composition and richness than maximum VPD or maximum temperature. At the same time, maximum temperature did not explain any additional variation in species composition and richness not explained by mean VPD. Our results thus provide strong evidence that the mean VPD is more relevant predictor of bryophyte community composition and richness than maximum temperature or maximum VPD. The unique effects of mean VPD, not reflected by the maximum temperature or maximum VPD, suggest that bryophyte communities are more sensitive to the long-term characteristics of site microclimatic conditions, rather than to short-term microclimatic extremes captured by maxima.

Second, our results showing that actual vapor pressure is relatively constant across the landscape imply that it is possible to estimate VPD from local microclimate air temperature measurements combined with non-local measurements of air relative humidity, for example from a nearby weather station. While the general applicability of this approach should be further tested across spatial scales (Dahlberg et al., 2020), in various environmental settings and different vegetation types, our findings suggest that local VPD can be reasonably estimated (Appendix B, Fig. B1). This finding thus opens exciting possibilities for further research as local temperature measurements are increasingly available all over the world (Lembrechts et al., 2020). However, it should be stressed that this

approach generates VPD estimates which provide reasonable ranking of the sites along the VPD gradient, but generally overestimate the VPD (Appendix B, Fig. B1), likely because it does not account for locally higher actual vapor pressure, for example near springs, water bodies or on permanently waterlogged soils.

### 4.1 VPD variability across the landscape

358359

360361

363364

373374

Large spatial variability in atmospheric VPD structured forest bryophyte communities across the landscape. Interestingly, VPD variation was driven by temperature-controlled P<sub>sat</sub>, while P<sub>air</sub> was relatively constant across the landscape. This finding is important, as the actual vapor pressure should also be variable across the landscape (Ogeé et al. 2024; Johnston et al., 2025). However, our findings suggest that the local and spatially highly heterogeneous processes like evaporation from soil and water surfaces and plant transpiration contribute little to the landscape-scale variation in VPD, even in the topographically diverse landscape with steep microclimatic gradients.

While maximum VPD was solely driven by saturated vapor pressure and therefore maximum temperature, the mean VPD was more affected by actual vapor pressure. However, saturated and actual vapor pressures were negatively correlated and therefore the unique effect of actual vapor pressure on spatial pattern in atmospheric VPD was surprisingly small. The landscape-scale variation in atmospheric VPD was therefore controlled by microclimate temperature variation.

Microclimate temperature variation over the landscape, crucial for community ecology, is largely dictated by land-surface topography (Dobrowski, 2011). Land-surface topography controls also maximum air temperatures in the forest understory (Vanwalleghem and Meentemeyer, 2009; Macek et al., 2019) and therefore spatial variability in saturation vapor pressure. However, we were surprised that the highly localized processes like evapotranspiration did not contribute much to the spatial variability in absolute air humidity despite our study area with extremely rugged topography and contrasting forest vegetation types. Therefore, spatial variability in absolute air humidity seems to be determined mostly by processes operating at much larger scales like atmospheric circulation and precipitation patterns (Campbell and Norman, 1998). Nevertheless, local topographic depression with waterlogged soils and especially the proximity to flowing water or permanent water bodies can locally elevate actual vapor pressure and therefore decrease atmospheric VPD (Wei et al. 2018, Ogeé et al. 2024) However, our results suggest that the overall pattern in atmospheric VPD will generally follow changes in air temperature and therefore future climate warming will result in non-linear increase in evaporative stress across the landscapes. Given the growing recognition of VPD importance for many ecosystem processes, plant distribution, and community assembly (Grossiord et al., 2020; Kopecký et al., 2024; Novick et al., 2024), the approach we developed here to disentangle the contribution of saturated versus actual vapor pressure can provide new insights into the drivers of VPD variability across spatial and temporal scales. So far, the knowledge of the relative importance of saturated versus actual vapor pressure is limited, therefore it is difficult to compare our results with other studies. Nevertheless, a comparison of the drivers of VPD variability across agricultural fields in Germany supports our conclusion that temperature-driven variability in saturated vapor pressure is a dominant control of VPD variability at finer scales (Wörlen et al., 1999).

### 4.2 VPD effects on bryophytes

412

413

414

In contrast to vascular plants, bryophytes tolerate desiccation and become metabolically inactive in the absence of 378 water (Proctor, 2000). When conditions improve, bryophytes quickly reactivate physiological processes such as 379 respiration, photosynthesis, cell cycle, or normal cytoskeleton function (Proctor, Ligrone, et al., 2007; Proctor, 380 Oliver, et al., 2007). However, this reactivation requires a lot of energy, for example to produce specific repair 381 proteins (Oliver and Bewley, 1984; Zeng et al., 2002) or to maintain the integrity and normal function of cell 382 organelles and membranes (Platt et al., 1994). Prolonged periods without evaporative stress are therefore key for bryophyte growth and long-term survival (Proctor, Oliver, et al., 2007; Merinero et al., 2020). 383 384 Bryophyte cells at full turgor have osmotic potential rarely more negative than -2 MPa (Proctor, 2000). An osmotic potential of -1.36 MPa is in equilibrium with air at 20 °C and 99% relative humidity (i.e. VPD

Figure A1: Pearson correlation matrix of microclimatic variables representing evaporative stress (maximum temperature, maximum and mean VPD) and its components (mean P<sub>sat</sub> and mean P<sub>air</sub>).

# Appendix B

# VPD estimate from in-situ air temperature and regional air humidity

Based on our results, we speculated that local atmospheric VPD can be reasonably estimated using the in-situ air temperature measurements paired with relative air humidity measurements representative for the whole region (and therefore the same for all plots situated within that region).

To explore this idea, we estimated the mean VPD using in-situ measured air temperature (HOBO U23 ProV2 dataloggers in 1.5 m height) and relative air humidity measured in the Tokáň weather station located in the study area (Fig. 1).

While the measured and estimated VPD were closely correlated (Pearson R= 0.97), estimated VPD were consistently higher than in-situ measured VPD (Fig. B1).

Therefore, we conclude that the relative position of the site on the VPD gradient can be reasonably estimated from in-situ microclimate temperature measurements paired with regional relative air humidity measurements. However, it should be stressed that this approach generates VPD estimates which provide reasonable ranking of the sites along the VPD gradient, but generally overestimate the VPD (Fig. B1), likely because it does not account for locally higher actual vapor pressure, for example near springs, water bodies or on permanently waterlogged soils. Therefore, this approach cannot fully replace local air humidity measurements.

Figure B1: Relationship between in-situ measured mean VPD and mean VPD estimated from in-situ measured air temperature and relative air humidity measured in regional weather station (June-August 2022). While the measured and estimated VPD are closely correlated (Pearson R=0.97), estimated VPD tends to be higher than in-situ measured VPD, likely because of locally higher air humidity in topographically sheltered sites near valley bottoms.

# Appendix C

# List of bryophyte species

Table C1: Complete species list of bryophyte species recorded at 38 study plots.

| Species name               | Occurence | Taxonomic group |
|----------------------------|-----------|-----------------|
| 1 Dicranum scoparium       | 32        | moss            |
| 2 Leucobryum juniperoideum | 26        | moss            |
| 3 Hypnum cupressiforme     | 24        | moss            |
| 4 Tetraphis pellucida      | 21        | moss            |
| 5 Bazzania trilobata       | 18        | liverwort       |
| 6 Polytrichum formosum     | 17        | moss            |

| 7  | Chiloscyphus profundus              | 15 | liverwort |
|----|-------------------------------------|----|-----------|
| 8  | Plagiothecium laetum/curvifolium    | 15 | moss      |
| 9  | Orthodontium lineare                | 13 | moss      |
| 10 | Plagiothecium undulatum             | 11 | moss      |
| 11 | Pleurozium schreberi                | 10 | moss      |
| 12 | Sphagnum girgensohnii/capillifolium | 10 | moss      |
| 13 | Dicranodontium denudatum            | 9  | moss      |
| 14 | Campylopus flexuosus                | 8  | moss      |
| 15 | Lepidozia reptans                   | 8  | liverwort |
| 16 | Chiloscyphus cuspidatus             | 8  | liverwort |
| 17 | Pohlia nutans                       | 8  | moss      |
| 18 | Mnium hornum                        | 7  | moss      |
| 19 | Calypogeia integristipula           | 6  | liverwort |
| 20 | Herzogiella seligeri                | 5  | moss      |
| 21 | Brachythecium rutabulum             | 4  | moss      |
| 22 | Calypogeia mulleriana               | 4  | liverwort |
| 23 | Dicranella heteromalla              | 4  | moss      |
| 24 | Orthodicranum montanum              | 4  | moss      |
| 25 | Mylia taylorii                      | 3  | liverwort |
| 26 | Atrichum undulatum                  | 2  | moss      |
| 27 | Dicranum majus                      | 2  | moss      |
| 28 | Odontoschisma denudatum             | 2  | liverwort |
| 29 | Pellia epiphylla                    | 2  | liverwort |
| 30 | Polytrichum commune                 | 2  | moss      |
| 31 | Ptilidium ciliare                   | 2  | liverwort |
| 32 | Cephalozia bicuspidata              | 1  | liverwort |
| 33 | Dicranoweisia cirrata               | 1  | moss      |
| 34 | Lophozia ventricosa                 | 1  | liverwort |
| 35 | 9 46                                | 1  | moss      |
| 36 | 8                                   | 1  | moss      |
| 37 | į č                                 | 1  | moss      |
| 38 | •                                   | 1  | liverwort |
| 39 | Scapania nemorea                    | 1  | liverwort |
|    |                                     |    |           |

Data availability. The data supporting the findings of this study are currently provided on GitHub public repository
 (https://doi.org/10.5281/zenodo.15805801).

- Author contribution. Conceptualization: AR, MMan, MMac, JW, MK. Funding acquisition: MK. Data curation:
- AR. Methodology: MMac, MK. Formal analysis: AR. Investigation: AR, MMan, MMac, JW, MK. Visualization:
- AR, MMan, MMac, MK. Writing original draft: AR. Writing review & editing: AR, MMan, MMac, JW, MK.
- Supervision: MK.
- Competing interest. The authors declare that they have no conflict of interest.
- Acknowledgements. We thank all colleagues who helped us to collect microclimate data, the Administration of
- the Bohemian Switzerland National Park for the long-term support, and Caroline Greiser, Alain Vanderpoorten
- and anonymous reviewer for their useful comments and suggestions.
- Financial support. This study was supported by the Czech Science Foundation (project GACR 23-06614S)
- and the Czech Academy of Sciences (project RVO 67985939).

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

Cretaceous Basin (Czech Republic/Germany/Poland), in: Sandstone landscapes, edited by:

| 559 | Härtel, H., Cílek, V., Herben, T., Jackson, A., and Williams, R., Academia, Praha, 177-189,                                                                        |
|-----|--------------------------------------------------------------------------------------------------------------------------------------------------------------------|
| 560 | https://doi.org/10.6084/m9.figshare.92598, 2007.                                                                                                                   |
| 561 | Hearnshaw, G. F. and Proctor, M. C. F.: The effect of temperature on the survival of dry bryophytes, New                                                           |
| 562 | Phytol., 90(2), 221–228, https://doi.org/10.1111/j.1469-8137.1982.tb03254.x, 1982.                                                                                 |
| 563 | Hill, M. O. and Preston, C. D.: The geographical relationships of British and Irish bryophytes, J. Bryol., 20(1)                                                   |
| 564 | 127–226, https://doi.org/10.1179/jbr.1998.20.1.127, 1998.                                                                                                          |
| 565 | Hinshiri, H. M. and Proctor, M. C. F.: The effect of desiccation on subsequent assimilation and respiration of                                                     |
| 566 | the bryophytes Anomodon viticulosus and Porella platyphylla, New Phytol., 70(3), 527-538,                                                                          |
| 567 | https://doi.org/10.1111/j.1469-8137.1971.tb02554.x, 1971.                                                                                                          |
| 568 | Hjort, J., Heikkinen, R. K., and Luoto, M.: Inclusion of explicit measures of geodiversity improve biodiversity                                                    |
| 569 | models in boreal landscape, Biodivers. Conserv., 21, 3487-3506,                                                                                                    |
| 570 | https://doi.org/10.1007/s10531-012-0376-1, 2012.                                                                                                                   |
| 571 | IPCC: Climate Change 2023: synthesis report, <a href="https://doi.org/10.59327/IPCC/AR6-9789291691647">https://doi.org/10.59327/IPCC/AR6-9789291691647</a> , 2023. |
| 572 | Johnston, M. R, Barnes, M. L., Preisler, Y., Smith, W. K., Biederman, J. A., Scott, R. L., Williams, A. P., and                                                    |
| 573 | Dannenberg, M. P.: Effects of hot versus dry vapor pressure deficit on ecosystem carbon and                                                                        |
| 574 | water fluxes, J. Geophys. ResBiogeo., 130(1), e2024JG008146,                                                                                                       |
| 575 | https://doi.org/10.1029/2024JG008146, 2025.                                                                                                                        |
| 576 | Jones, H. G.: Plants and microclimate: a quantitative approach to environmental plant physiology, 3rd ed.,                                                         |
| 577 | Cambridge University Press, <a href="https://doi.org/10.1017/CBO9780511845727">https://doi.org/10.1017/CBO9780511845727</a> , 2014.                                |
| 578 | Kopecký, M., Hederová, L., Macek, M., Klinerová, T., and Wild, J.: Forest plant indicator values for moisture                                                      |
| 579 | reflect atmospheric vapour pressure deficit rather than soil water content, New Phytol., 244(5),                                                                   |
| 580 | 1801-1811., https://doi.org/10.1111/nph.20068, 2024.                                                                                                               |
| 581 | Kučera, J., Váňa, J., and Hradílek. Z.: Bryophyte flora of the Czech Republic: updated checklist and Red List                                                      |
| 582 | and a brief analysis, Preslia, 84(3), 813-850, 2012.                                                                                                               |
| 583 | Lawrence, M. G.: The relationship between relative humidity and the dewpoint temperature in moist air: a                                                           |
| 584 | simple conversion and applications, B. Am. Meteorol. Soc., 86(2), 225-234,                                                                                         |
| 585 | https://doi.org/10.1175/BAMS-86-2-225, 2005.                                                                                                                       |
| 586 | Legendre, P.: Studying beta diversity: ecological variation partitioning by multiple regression and canonical                                                      |
| 587 | analysis, J. Plant Ecol.,1(1), 3–8, https://doi.org/10.1093/jpe/rtm001, 2008.                                                                                      |
| 588 | Legendre, P., Oksanen, J., and ter Braak, C. J. F.: Testing the significance of canonical axes in redundancy                                                       |
| 589 | analysis, Methods Ecol. Evol., 2(3), 269–277, https://doi.org/10.1111/j.2041-                                                                                      |
| 590 | <u>210X.2010.00078.x</u> , 2011.                                                                                                                                   |
| 591 | Lembrechts, J. J., Aalto, J., Ashcroft, M. B., et al.: SoilTemp: a global database of near-surface temperature,                                                    |
| 592 | Glob. Change Biol., 26(11), 6616–6629, https://doi.org/10.1111/gcb.15123, 2020.                                                                                    |
| 593 | León-Vargas, Y., Engwald, S., and Proctor, M. C. F.: Microclimate, light adaptation and desiccation tolerance                                                      |
| 594 | of epiphytic bryophytes in two Venezuelan cloud forests, J. Biogeogr., 33(5), 901-913,                                                                             |
| 595 | https://doi.org/10.1111/j.1365-2699.2006.01468.x, 2006.                                                                                                            |
| 596 | López, J., Way, D. A., and Sadok, W.: Systemic effects of rising atmospheric vapor pressure deficit on plant                                                       |
| 597 | physiology and productivity, Glob. Change Biol., 27(9), 1704-1720,                                                                                                 |
| 598 | https://doi.org/10.1111/gcb.15548, 2021.                                                                                                                           |

- Lösch, R., Kappen, L., and Wolf, A.: Productivity and temperature biology of two snowbed bryophytes, Polar 600 Biol., 1(4), 243-248, https://doi.org/10.1007/BF00443195, 1983. Lu, H., Qin, Z., Lin, S., Chen, X., Chen, B., He, B., Wei, J., and Yuan, W.: Large influence of atmospheric 601 vapor pressure deficit on ecosystem production efficiency, Nat. Commun., 13(1), 10–13, 602 603 https://doi.org/10.1038/s41467-022-29009-w, 2022. 604 Macek, M., Kopecký, M., and Wild, J.: Maximum air temperature controlled by landscape topography affects 605 plant species composition in temperate forests, Landscape Ecol., 34, 2541-2556, https://doi.org/10.1007/s10980-019-00903-x, 2019. 606 607 Máliš, F., Ujházy, K., Hederová, L., Ujházyová, M., Csölleová, L., Coomes, D. A., and Zellweger, F.: 608 Microclimate variation and recovery time in managed and old-growth temperate forests, Agr. 609 Forest Meteorol., 342, 109722, https://doi.org/10.1016/j.agrformet.2023.109722, 2023. Man, M., Kalčík, V., Macek, M., Brůna, J., Hederová, L., Wild, J., and Kopecký, M.: myClim: microclimate 610 611 data handling and standardised analyses in R, Methods Ecol. Evol., 14(9), 2308–2320, 612 https://doi.org/10.1111/2041-210X.14192, 2023. 613 Man, M., Wild, J., Macek, M., and Kopecký, M.: Can high-resolution topography and forest canopy structure 614 substitute microclimate measurements? Bryophytes say no., Sci. Total Environ., 821, 153377, https://doi.org/10.1016/j.scitotenv.2022.153377, 2022. 615 616 Marková, I.: Mechorosty Českého Švýcarska (Labských pískovců), in: Labské pískovce - historie, příroda a ochrana území, edited by: Bauer, P., Kopecký, V., and Šmucar, J., Agentura ochrany přírody 617 a krajiny ČR, Správa CHKO Labské pískovce, Děčín, 106-120, 2008. [in Czech language] 618 619 Merinero, S., Dahlberg, C. J., Ehrlén, J., and Hylander, K.: Intraspecific variation influences performance 620 of moss transplants along microclimatic gradients, Ecology, 101(5), e02999, 621 https://doi.org/10.1002/ecy.2999, 2020. 622 McArdle, B. H. and Anderson, M. J.: Fitting multivariate models to community data: a comment on distance-623 based redundancy analysis, *Ecology*, 82(1), 290–297, https://doi.org/10.1890/0012-624 9658(2001)082[0290:FMMTCD]2.0.CO;2, 2001. 625 McDowell, N., Pockman, W. T., Allen, C. D., Breshears, D. D., Cobb, N., Kolb, T., Plaut, J., Sperry, J., West, A., Williams, D. G., and Yepez, E. A.: Mechanisms of plant survival and mortality during 626 627 drought: why do some plants survive while others succumb to drought?, New Phytol., 178(4), 719-739, https://doi.org/10.1111/j.1469-8137.2008.02436.x, 2008. 628 Morales-Sánchez, J. Á. M., Mark, K., Souza, J. P.S., and Niinemets, Ü.: Desiccation - rehydration 629 630 measurements in bryophytes: current status and future insights, J. Exp. Bot., 73(13), 4338–4361, 631 https://doi.org/10.1093/jxb/erac172, 2022. Novick, K. A., Ficklin, D. L., Grossiord, C., Konings, A. G., Martínez-Vilalta, J., Sadok, W., Trugman, A. T., 632 633 Williams, A. P., Wright, A. J., Abatzoglou, J. T., Dannenberg, M. P., Gentine, P., Guan, K., Johnston, M. R., Lowman, L. E. L., Moore, D. J. P., and McDowell, N. G.: The impacts of 634 635 rising vapour pressure deficit in natural and managed ecosystems, Plant Cell Environ., 47(9), 3561-3589, https://doi.org/10.1111/pce.14846, 2024. 636
  - Novick, K. A., Ficklin, D. L., Stoy, P. C., Williams, C. A., Bohrer, G., Oishi, A. C., Papuga, S. A., Blanken, P. D., Noormets, A., Sulman, B. N., Scott, R. L., Wang, L., and Phillips, R. P.: The increasing

637

| 639 | importance of atmospheric demand for ecosystem water and carbon fluxes, Nat. Clim. Change,                                                                   |
|-----|--------------------------------------------------------------------------------------------------------------------------------------------------------------|
| 640 | 6(11), 1023–1027, <a href="https://doi.org/10.1038/nclimate3114">https://doi.org/10.1038/nclimate3114</a> , 2016.                                            |
| 641 | Ogée, J., Walbott, M., Barbeta, A., Corcket, E., and Brunet, Y.: Decametric-scale buffering of climate                                                       |
| 642 | extremes in forest understory within a riparian microrefugia: the key role of microtopography,                                                               |
| 643 | Int. J. Biometeorol., 68(9), 1741–1755, <a href="https://doi.org/10.1007/s00484-024-02702-9">https://doi.org/10.1007/s00484-024-02702-9</a> , 2024.          |
| 644 | Oksanen, J., Simpson, G., Blanchet, F., Kindt, R., Legendre, P., Minchin, P. R., O'Hara, R. B., Solymos, P.,                                                 |
| 645 | Stevens, M. H. H., Szoecs, E., Wagner, H., Barbour, M., Bedward, M., Bolker, B., Borcard, D.,                                                                |
| 646 | Carvalho, G., Chirico, M., De Caceres, M., Durand, S., Evangelista, H. B. A., FitzJohn, R.,                                                                  |
| 647 | Friendly, M., Furneaux, G., Hill, M. O., Lahti, L., McGlinn, D., Ouellette, M-H., Cunha, E. R.,                                                              |
| 648 | Smith, T., Stier, A., ter Braak, C. J. F., Weedon, J., and Borman, T.: vegan: Community                                                                      |
| 649 | Ecology Package [R package vegan version 2.6-4], <a href="https://cran.r-project.org/package=vegan">https://cran.r-project.org/package=vegan</a> ,           |
| 650 | 5 Feb. 2025, 2022.                                                                                                                                           |
| 651 | Oliver, M. J. and Bewley, J. D.: Plant desiccation and protein synthesis, Plant Physiol., 74(4), 923-927,                                                    |
| 652 | https://doi.org/10.1104/pp.74.4.923, 1984.                                                                                                                   |
| 653 | Oliver, M. J., Velten, J., and Wood, A. J.: Bryophytes as experimental models for the study of environmental                                                 |
| 654 | stress tolerance: Tortula ruralis and desiccation-tolerance in mosses, Plant Ecol., 151(1), 73-84.                                                           |
| 655 | https://doi.org/10.1023/A:1026598724487, 2000.                                                                                                               |
| 656 | Pardow, A. and Lakatos, M.: Desiccation tolerance and global change: implications for tropical bryophytes in                                                 |
| 657 | lowland forests, Biotropica, 45(1), 27–36, <a href="https://doi.org/10.1111/J.1744-7429.2012.00884.X">https://doi.org/10.1111/J.1744-7429.2012.00884.X</a> , |
| 658 | 2013.                                                                                                                                                        |
| 659 | Pedersen, T. and Crameri, F.: scico: Colour palettes based on the scientific colour maps [R package scio version                                             |
| 660 | 1.5.0], https://CRAN.R-project.org/package=scico, 11 Mar. 2025, 2023.                                                                                        |
| 661 | Peres-Neto, P. R., Legendre, P., Dray, S., and Borcard D.: Variation partitioning of species data matrices                                                   |
| 662 | estimation and comparison of fractions, Ecology, 87, 2614-25, https://doi.org/10.1890/0012                                                                   |
| 663 | 9658(2006)87[2614:VPOSDM]2.0.CO;2, 2006.                                                                                                                     |
| 664 | Platt, K. A., Oliver, M. J., and Thomson, W. W.: Membranes and organelles of dehydrated Selaginella and                                                      |
| 665 | Tortula retain their normal configuration and structural integrity: freeze fracture evidence,                                                                |
| 666 | Protoplasma, 178(1–2), 57–65, <a href="https://doi.org/10.1007/BF01404121">https://doi.org/10.1007/BF01404121</a> , 1994.                                    |
| 667 | Proctor, M. C. F. The bryophyte paradox: tolerance of dessication, evasion of drought, Plant Ecol., 151(1),                                                  |
| 668 | 41–49, <a href="https://doi.org/10.1023/A:1026517920852">https://doi.org/10.1023/A:1026517920852</a> , 2000.                                                 |
| 669 | Proctor, M. C. F. Patterns of desiccation tolerance and recovery in bryophytes, Plant Growth Regul., 35(2),                                                  |
| 670 | 147–156, <a href="https://doi.org/10.1023/A:1014429720821">https://doi.org/10.1023/A:1014429720821</a> , 2001.                                               |
| 671 | Proctor, M. C. F., Ligrone, R., and Duckett, J. G.: Desiccation tolerance in the moss <i>Polytrichum formosum</i> :                                          |
| 672 | physiological and fine-structural changes during desiccation and recovery, Ann. BotLondon,                                                                   |
| 673 | 99(1), 75–93, <a href="https://doi.org/10.1093/aob/mcl246">https://doi.org/10.1093/aob/mcl246</a> , 2007.                                                    |
| 674 | Proctor, M. C. F, Oliver, M. J., Wood, A. J., and Alpert, P.: Desiccation-tolerance in bryophytes: a review,                                                 |
| 675 | Bryologist, 110(4), 595–621, https://doi.org/10.1639/0007-                                                                                                   |
| 676 | 2745(2007)110[595:DIBAR]2.0.CO;2, 2007.                                                                                                                      |
| 677 | R Core Team: R: A language and environment for statistical computing, R foundation for statistical                                                           |
| 678 | computing, Vienna, Austria, <a href="https://www.R-project.org/">https://www.R-project.org/</a> , 15 Mar. 2025, 2024.                                        |

Rambo, T. R. and Muir, P. S.: Forest floor bryophytes of Pseudotsuga menziesii-Tsuga heterophylla stands in 680 Oregon: influence of substrate and overstory, Bryologist, 101(1), 116–130, https://doi.org/10.2307/3244083, 1998. 681 682 Rice, S. K., Collins, D., and Anderson, A. M.: Functional significance of variation in bryophyte canopy 683 structure, Am. J. Bot., 88(9), 1568–1576, https://doi.org/10.2307/3558400, 2001. Ruehr, N. K., Law, B. E., Quandt, D., and Williams, M.: Effects of heat and drought on carbon and water 684 685 dynamics in a regenerating semi-arid pine forest: a combined experimental and modelling 686 approach, Biogeosciences, 11, 4139-4156, https://doi.org/10.5194/bg-11-4139-2014, 2014. 687 Schmalholz, M. and Hylander, K.: Microtopography creates small-scale refugia for boreal forest floor 688 bryophytes during clear-cut logging, Ecography, 34(4), 637–348, https://doi.org/10.1111/j.1600-0587.2010.06652.x, 2011. 689 690 Schofield, W. B.: Ecological significance of morphological characters in the moss gametophyte, Bryologist, 84(2), 149–165, https://doi.org/10.2307/3242819, 1981. 691 692 Schönbeck, L. C., Schuler, P., Lehmann, M. M., Mas, E., Mekarni, L., Pivovaroff, A. L., Turberg, P., and 693 Grossiord, C.: Increasing temperature and vapour pressure deficit lead to hydraulic damages in 694 the absence of soil drought, Plant, Cell Environ., 45(11), 3275–3289, 695 https://doi.org/10.1111/pce.14425, 2022. 696 Sonnleitner, M., Dullinger, S., Wanek, W., and Zechmeister, H.: Microclimatic patterns correlate with the 697 distribution of epiphyllous bryophytes in a tropical lowland rain forest in Costa Rica, J. Trop. 698 Ecol., 25(3), 321–330, https://doi.org/10.1017/S0266467409006002, 2009. 699 Staude, I. R., Waller, D. M., Bernhardt-Römermann, M., Bjorkman, A. D., Brunet, J., De Frenne, P., Hédl, R., 700 Jandt, U., Lenoir, J., Máliš, F., Verheyen, K., Wulf, M., Pereira, H. M., Vangansbeke, P., 701 Ortmann-Ajkai, A., Pielech, R., Berki, I., Chudomelová, M., Decocq, G., Dirnböck, T., Durak, 702 T., Heinken, T., Jaroszewicz, B., Kopecký, M., Macek, M., Malicki, M., Naaf, T., Nagel, T. A., 703 Petřík, P., Reczyńska, K., Schei, F. H., Schmidt, W., Standovár, T., Świerkosz, K., Teleki, B., 704 Van Calster, H., Vild, O., and Baeten, L.: Replacements of small- by large-ranged species scale 705 up to diversity loss in Europe's temperate forest biome, Nature, 4, 802–808, 706 https://doi.org/10.1038/s41559-020-1176-8, 2020. 707 Tetens, O.: Ueber einige meteorologische Begriffe. Zeitschrift für geophysik, 6, 297-309, 1930. [in German 708 language] 709 Ulrich, W., Almeida-Neto, M., and Gotelli, N. J.: A consumer's guide to nestedness analysis, Oikos, 118, 3-710 17, https://doi.org/10.1111/j.1600-0706.2008.17053.x, 2009. 711 Vanderpoorten, A. and Engels, P.: The effects of environmental variation on bryophytes at regional scale, 712 Ecography, 25(5), 513-522, https://doi.org/10.1034/j.1600-0587.2002.250501.x, 2002. 713 Vanderpoorten, A. and Goffinet, B.: Introduction to bryophytes, Cambridge University Press, Cambridge, 303 714 pp., https://doi.org/10.1017/CBO9780511626838, 2009. 715 Vanwalleghem, T. and Meentemeyer, R. K.: Predicting forest microclimate in heterogeneous landscapes, 716 Ecosystems, 12, 1158–1172, https://doi.org/10.1007/s10021-009-9281-1, 2009. 717 Wagner, D.J. and Titus, J. E.: Comparative desiccation tolerance of two Sphagnum mosses, Oecologia, 62,

182–187, https://doi.org/10.1007/BF00379011, 1984.

719 Wei, L., Zhou, H., Link, T. E., Kavanagh, K. L., Hubbart, J. A., Du, E., Hudak, A. T., and Marshall, J. D.: 720 Forest productivity varies with soil moisture more than temperature in a small montane 721 watershed, Agr. Forest Meteorol., 259, 211-221, https://doi.org/10.1016/j.agrformet.2018.05.012, 2018. 722 Wild, J., Macek, M., Kopecký, M., Zmeškalová, J., Hadincová, V., and Trachtová, P.: Temporal and spatial 723 variability of microclimate in sandstone landscape: detailed field measurement, in: Proceedings 724 725 of the 3rd International Conference on Sandstone Landscapes, Sandstone Landscapes, Diversity, Ecology and Conservation, University of Wroclaw, 220–224, 2013. 726 Will, R. E., Wilson, S. M., Zou, C. B., and Hennessey, T. C.: Increased vapor pressure deficit due to higher 727 728 temperature leads to greater transpiration and faster mortality during drought for tree seedlings 729 common to the forest-grassland ecotone, New Phytol., 200(2), 366-374, 730 https://doi.org/10.1111/nph.12321, 2013. 731 Williams, A. P., Allen, C. D., Macalady, A. K., Griffin, D., Woodhouse, C. A., Meko, D. M., Swetnam, T. W., Rauscher, S. A., Seager, R., Grissino-Mayer, H. D., Dean, J. S., Cook, E. R., Gangodagamage, 732 733 C., Cai, M., and McDowell, N. G.: Temperature as a potent driver of regional forest drought 734 stress and tree mortality, Nat. Clim. Change, 3(3), 292–297, https://doi.org/10.1038/nclimate1693, 2013. 735 736 Wolf, K. D., Higuera, P. E., Davis, K. T., and Dobrowski, S. Z.: Wildfire impacts on forest microclimate vary with biophysical context, *Ecosphere*, 12(5), e03467, <a href="https://doi.org/10.1002/ecs2.3467">https://doi.org/10.1002/ecs2.3467</a>, 2021. 737 738 Wood, S. N.: Fast stable restricted maximum likelihood and marginal likelihood estimation of semiparametric 739 generalized linear models, J. R. Stat. Soc.: Series B (Statistical Methodology), 73(1), 3–36, 740 https://doi.org/10.1111/J.1467-9868.2010.00749.X, 2011. 741 Wörlen, C., Schulz, K., Huwe, B., and Eiden, R.: Spatial extrapolation of agrometeorological variables, Agr. For. Meteorol., 94(3-4), 233-242, <a href="https://doi.org/10.1016/S0168-1923(99)00015-5">https://doi.org/10.1016/S0168-1923(99)00015-5</a>, 1999. 742 743 Wright, D. H., Patterson, B. D., Mikkelson, G. M., Cutler, A., and Atmar, W.: A comparative analysis of 744 nested subset patterns of species composition, Oecologia, 113, 1–20, 745 https://doi.org/10.1007/s004420050348, 1997. Yuan, W., Zheng, Y., Piao, S., Ciais, P., Lombardozzi, D., Wang, Y., Ryu, Y., Chen, G., Dong, W., Hu, Z., 746 747 Jain, A. K., Jiang, C., Kato, E., Li, S., Lienert, S., Liu, S., Nabel, J. E. M. S., Qin, Z., Quine, T., Sitch, S., Smith, W. K., Wang, F., Wu, C., Xiao, Z., and Yang, S.: Increased atmospheric vapor 748 749 pressure deficit reduces global vegetation growth, Sci. Adv., 5(8), 1–13, 750 https://doi.org/10.1126/sciadv.aax1396, 2019. 751 Zeng, Q., Chen, X., Wood, A. J.: Two early light-inducible protein (ELIP) cDNAs from the resurrection plant 752 Tortula ruralis are differentially expressed in response to desiccation, rehydration, salinity, and 753 high light, J. Exp. Bot., 53(371), 1197–1205, https://doi.org/10.1093/jexbot/53.371.1197, 2002.