# Peer review of "Temperature-driven vapor pressure deficit structures forest"

_EGUsphere, 2025_

## Author Comment (AC1)

**Response RC1: Alain Vanderpoorten**

Dear Alain Vanderpoorten,

We would like to thank you very much for your detailed and constructive feedback on our manuscript *Temperature-driven vapor pressure deficit structures forest bryophyte communities across the landscapes*. We are grateful for your positive assessment of our manuscript and for appreciating the novelty and timing of our work. We will do our best to incorporate your helpful suggestions to make our manuscript clearer and more interesting for the readers.

In the following response, your reviewer's comments are written in standard black font, and our responses are written in blue.

Sincerely, on behalf of our author team,

Anna Růžičková

I found this paper most original and useful as it addresses the very timely question of microclimates on diversity patterns, with a special emphasis on one key parameter: VPD. To give the paper the impact that it deserves and emphasize the relevance of VPD as an important, ecologically meaningful variable, I have two suggestions to make: (i) define VPD and explain, in the Introduction, what its ecological relevance as compared to other microclimatic variables such as T or RH alone (in other words, why would it be important to integrate VPD in ecological studies) and (ii) show that VPD is indeed a better predictor of species richness and composition than T or RH alone. This could be easily done by re-running the analyses, using T and RH as predictors, and showing that the use of VPD results in a higher percent variance of bryophyte richness and composition community explained.

Regarding your first suggestion (i), we agree, and we will add a definition of VPD to the Introduction, as well as an explanation of its ecological relevance compared to other microclimatic variables. Our original aim was to make the Introduction as concise as possible and to focus directly on the VPD. But we agree with the reviewer that it is useful to emphasize and explain in more depth the reasons why it is important to integrate VPD into ecological studies instead of, e.g. the temperature itself or relative humidity.

Regarding the second suggestion (ii) - explanatory effects of VPD compared to T and RH. These three variables are closely corelated in our study area (see correlation matrix below). Therefore, we cannot statistically separate their effects. We agree that this is important information, which was not fully presented in the previous paper version. Therefore, we will add this information to the revised paper. In the paper, we focused directly on VPD, because it is the most physiologically relevant variable (as we discuss on lines 239-255 and 270-287), and because it integrates both temperature and air humidity in a theoretically and physically preferable way (Anderson 1936). However, we acknowledge that it is difficult to separate the effects of these variables with observation data collected on landscape scale and we will expand the discussion of this topic in the revision.

Because of the very tight correlation between VPD, T and RH in our study (see correlation matrix below), using these variables in analyses resulted in similar proportion of explained variability (see table below). Nevertheless, VPD consistently explained more variability in species composition than either T or RH. For species richness, VPD explained more variability than RH, but slightly less than T. However, for reasons given above (e.g. physiological relevance), we prefer to focus on VPD and clearly acknowledge that the VPD is closely related to the temperature within our study region. Of course, it would be interesting to separate effects of T and VPD, but this would require more extensive dataset collected across larger area or preferably experimental approach (as we already discuss on lines 203-209).

Nevertheless, while thinking about this comment from the reviewer, we realized that we should also explore not only the effects of the maximum VPD, but also average VPD. We initially decided to use the maximum VPD, because maximum temperature was the most important microclimatic variable affecting species composition of vascular plants (Macek et al., 2019) as well as bryophytes (Man et al., 2022). However, thanks to the reviewer's second comment (ii), we have renewed the discussion in our author team, and we have decided to reanalyze our data with the average VPD. The main motivation for this change was that the averages better capture the long-term characteristics of site microclimatic conditions, which are likely more important for bryophytes than short-term extremes captured by the maximum VPD (which can however be more relevant for vascular plants as they lack bryophytes unique ability to tolerate desiccation).

Using the average VPD instead of maximum VPD resulted in substantial increase in the explained variation, both in bryophyte community composition as well as species richness (see table below). Therefore, we decided to focus on mean VPD instead of maximum VPD in our revised paper. This change increases the proportion of explained variability in bryophyte species composition and species richness, but did not affect other conclusions about the importance of temperature as the main driver of VPD variability across the landscape.

[Figure]

*Correlation matrix of used microclimatic variables (VPD – the average daily mean VPD; airT – the average daily mean T, Psat – the average daily mean $P_{sat}$, Pair – the average daily mean $P_{air}$; RH – the average daily mean RH.*

|  | Species composition (db-RDA) | | Species richness (GAM) |
|---|---|---|---|
|  | **Sørensen** | **Simpson** | **Number of species** |
| **VPD max** | 10.95 % ** | 13.52 % ** | 31.2 % *** |
| **VPD mean** | 16.09 % *** | 17.15 % *** | 32.8 % *** |
| **T$_{air}$ mean** | 15.36 *** | 13.67 ** | 36.7 % *** |
| **RH mean** | 15.09 *** | 16.09 *** | 27.2 % *** |

*Table with results of db-RDA and GAM models*

One of the first results being discussed is the tight relationship between VPD and Tmax-it does not seem to me that this results directly stems out the analyses presented? Could this relationship be actually evidenced based on the data generated?

Yes, based on our data, we have evidenced a close relationship between VPD and T. This result stems from the analyses presented (Results part 3.1 VPD variability, second paragraph). However, it is true that we do not directly report the relationship between VPD and T, but instead the relationship between VPD and saturation vapor pressure (P$_{sat}$), which is solely a function of temperature (as we stated on line 43 and 109), so the relationship between VPD and P$_{sat}$ also illustrates the relationship between VPD and T.
We agree that the result presented in this way, without re-emphasizing the relationship between P$_{sat}$ and T, can be a little bit unclear for the reader not familiar with VPD calculation. Therefore, we will explain this more clearly and add to the Appendix a plot directly showing the relationship between VPD and T (see below). To elucidate the results even further, we will show in the revised Figure 3 results of variation partitioning based on a multiple linear regression model with the average daily mean VPD as the response variable and the average daily mean P$_{sat}$ and P$_{air}$ as the predictors (see updated Figure 3 below).

[Figure]

[Figure]

*On the left scatterplot the average daily maximum VPD ~ the average daily maximum T; on the right scatterplot the average daily mean VPD ~ the average daily mean T*

[Figure]

*Updated Figure 3 shows the relationships between VPD and saturated (a) and actual (b) vapor pressure. Inset Venn diagram shows that the landscape-scale variation in VPD is dominantly controlled by saturated vapor pressure (based on the result from variation partitioning).*

In the meantime, if VPD is tightly correlated with Tmax, does this not slightly undermine the premises of the study, that is, the potential benefit of an integrative variable such as VPD in ecological studies as compared to a 'simple' variable like Tmax?

While we found here that in the temperate forest landscape, VPD is dominantly driven by saturated vapor pressure (respectively T), and the contribution of actual vapor pressure is minor, the situation at larger spatial scales or near water bodies may be different. Under these conditions, the role of actual vapor pressure may be more prominent, and T alone may no longer be a good proxy of VPD. Therefore, we cannot recommend the use of a "simple" variable such as T instead of VPD. Generally, VPD is more physiologically relevant than T (as we discuss within the manuscript) and even if we found that the T is closely correlated to VPD in our study area, this correlation can change under the different conditions elsewhere. Therefore, we would still recommend measuring VPD directly, but also acknowledge that on landscape and finer scales, gradients in VPD and temperature can be very closely correlated and therefore difficult to distinguish (as we discuss on lines 270-287).

Since the second main result discussed is that it is possible to estimate VPD from local T measurements with HR measured as nearby weather stations, I suggest moving the content of this appendix into the result section of the main text. Would that mean that one can efficiently characterize microclimates using temperature sensors only, which are much cheaper than sensors combining T+HR?

We considered including the estimation of VPD from local T measurements with air humidity measured at a nearby weather station in the main results, but we decided not to do it because this is more of an application of our results rather than a hypothesis we have had at the beginning of our research. For this reason, we would prefer to leave this section in the Appendix. However, we are prepared to move these results into the main text, if the editor prefers to do so.

The answer to the second part of your question, whether that would mean that one can efficiently characterize microclimates using temperature sensors, is partly discussed in our response to your previous question about using simple variables like T instead of VPD.  Our approach to the VPD estimation can indeed be useful in cases where air humidity measurements are not available, but this should be done carefully and should be more widely tested across different biotopes and spatial scales. Moreover, as we stated in the Appendix (lines 317-320), this approach does not provide a reliable estimate of

absolute local atmospheric VPD, but rather the relative position on the VPD gradient. So, we would not recommend characterizing microclimate using temperature sensors only.

I was a bit surprised by the relatively limited contribution of VPD (about 11%) to variation in species composition among plots, whereas the introduction rightly emphasizes that in poikilohydric organisms like bryophytes, one could expect VPD to be a prime factor driving community composition. Looking at Fig1, one would think that communities at the top of a cliff would be very different from those in buffered conditions. I wonder whether this could be due to the fact that, as Fig1 suggests, this is a very rugged terrain, and that there is hence a huge (intra-plot) micro-habitat variation that is actually more important than (inter-plot) microclimatic variation. More information on the sampling sites would be welcome to understand the spatial heterogeneity potentially present in the sampling plots.

Compositional variation explained by a multivariate analysis depends on a dataset internal heterogeneity and therefore it is not comparable among datasets (Økland, 1999). Especially in ecological studies covering different vegetation types with limited overlap of the species composition between plots (as in our case), the percentage of the compositional variation explained by one variable is typically in similar range (Økland, 1999). To put the percentage of the explained variation further into the perspective, the best theoretically possible variable can explain max. of 30 % (Sørensen dissimilarity), respectively 54 % (Simpson dissimilarity) of compositional variation in our dataset (for methodological explanation see e.g. Macek et al. 2019). We decided not to complicate the paper with the reporting of this maximum explainable compositional variability which will require additional description of the methods used. However, we are ready to add this information into the revised paper if the Editor think it will be useful for better understanding of our results.

Moreover, motivated by the reviewer comment about the relevance of maximum VPD, we recalculated our results with average VPD instead of maximum VPD, which resulted in the substantial increase in the compositional variation explained by our analyses (db-RDA with average VPD: 16 % explained (Sørenson dissimilarity), 17 % explained (Simpson dissimilarity). We are therefore confident that these results support our conclusion that atmospheric VPD is an important driver of bryophyte species composition across the studied landscape.

It would help the reader if it was reminded in the Result section based on which analysis each result was obtained. For example, the variation partitioning and db-RDA are not mentioned in the Result section, and mentioning them would help the reader making a link between the M&Ms and results. For example, I am not sure which analysis was implemented to reach the result described L179-180 ('ecological relevance of VPD as compared to HR alone').

In the results, we will clearly state which results are based on which analysis. Specifically, the results presented on lines 179-180 were obtained by the *envfit* function from the *vegan* R package with 999 random permutations (as described on lines 141-143).

In the discussion, it would be interesting to add a section explaining what could be the factors accounting for the spatial variation of VPD reported, and why Pair exhibits such a comparatively narrow range of variation. At present the discussion falls a bit short—especially since the entire §4.2 (from L239 onwards) actually belongs to the Introduction (why bryophytes would be sensitive indicators of VPD variation), and not to the Discussion as it does not help interpreting the results presented.

Thank you for this suggestion. We agree that section 4.2 in the Discussion belongs more to the Introduction. Therefore, we will move this section to the Introduction. Regarding the factors accounting for the spatial variation of VPD, we already discussed them in section 4.1 of the Discussion (lines 223-230). However, in the revised paper, we will expand this discussion even further.

**References**

Anderson, D. B.: Relative humidity or vapor pressure deficit, Ecology, 17(2), 277–282, https://doi.org/10.2307/1931468, 1936.

Macek, M., Kopecký, M., and Wild, J.: Maximum air temperature controlled by landscape topography affects plant species composition in temperate forests, Landscape Ecol., 34, 2541–2556, https://doi.org/10.1007/s10980-019-00903-x, 2019.

Man, M., Wild, J., Macek, M., and Kopecký, M.: Can high-resolution topography and forest canopy structure substitute microclimate measurements? Bryophytes say no., Sci. Total Environ., 821, 153377, https://doi.org/10.1016/j.scitotenv.2022.153377, 2022.

Økland, R.H.: On the variation explained by ordination and constrained ordination axes, J. Veg. Sci., 10, 131–136, https://doi.org/10.2307/3237168, 1999.

---

## Author Comment (AC2)

**Response RC2: Anonymous**

Dear authors,

I very much enjoyed reading your manuscript and think you have done a good job in both setting the question, conducting the research and writing the paper. It was easy to read and I think you have many strong points to raise including the strong effects of VPD, that it is the saturation that varies, the strong link to maximum temperature and what that implies for our possibility to monitor and study the effects of VPD. Moreover you have illustrated that well with a data set of bryophytes.

I will try to be constructive to point out a few things that perhaps could improve the clarity of the paper. In general I think you could take a careful look at the flow of the text to avoid repetition and increase clarity.

Dear reviewer,

We kindly thank you for the positive acceptance of our manuscript, and we are pleased that you enjoyed reading it. Thank you for your constructive approach and helpful feedback. We will do our best to incorporate your suggestions and improve the clarity and fluency of the text.

In the following response, your reviewer's comments are written in standard black font, and our responses are written in blue.

At the same time, we would like to draw your attention to a change we would like to make based on suggestions from reviewer RC1 – we would like to use the average microclimatic values instead of the maximal values. The reasons are discussed in our response to RC1.

Sincerely, on behalf of our author team,

Anna Růžičková

**Design of the study.**

There is always a trade-off related to the size of the plot to study. A small plot as yours is good to capture the microclimate at one spot, but you will miss a lot of rare species in the landscape. In your case you have a circular plot of 1 meter radius, right? If wo it could be good to spell out. I lack some information on how you selected sites. Was it done using maps and satellite images and getting a coordinate from there? How did you select them in the field? What if a tree was in the plot? Or a big boulder? Did you make any notes on substrate? Substrate composition is often an important driver of species composition of bryophytes. You have an ambitious approach of covering the whole forest landscapes and then perhaps your sample size of 38 plots is a bit low. But you got very interesting results and have an interesting approach so I am still fine with this.

Yes, the species data presented in our manuscript were recorded on a circular plot of 1 meter radius. These plots were selected through stratified-random sampling to capture the main microclimatic gradients within the core zone of the national park (lines 95-96). Specifically, using GIS and detailed digital terrain model, we first divided the core zone into geographical strata defined by the position on the terrain (valley bottoms, lower slopes, upper slopes and ridges) and further separated the slopes into the slopes with predominantly northern and southern orientation. Then, we used GIS algorithms to randomly sample the equal number of locations within each defined strata with the additional conditions that the sampled locations must be separated by at least 300 m.

In the field, we navigated to the selected location with GPS device and placed the center of the plot 1 m to the north from the nearest tree. This tree was later equipped with the HOBO datalogger for air temperature and humidity measurements. Additional condition for plot selection was that the circular

area with a 1 m radius around the plot center must not contain any rocks or big stones in order to reduce the within plot substrate heterogeneity.

We agree that it would be nice to have more plots with complete data (both in situ measured microclimate and sampled bryophytes). However, we think that the 38 plots used in this study is sufficient for our aims. As we described above, the plots were carefully selected through stratified-random sampling. Therefore, they provide representative sample of the environmental variability within the core zone of the national park. The potential effects of within plot substrate heterogeneity were further reduced by the additional criteria for the plot selection (specified above).

Regarding the size of the research plots - we discussed it a lot, because we also collected larger (100 m$^2$) plots in each measurement site. The smaller (3.14 m$^2$) plots were always nested in the center of the larger 100 m$^2$ plot. However, during the bryophyte sampling, we did not make detailed records of the substrate, so we finally decided to based our analyses on the smaller plots (3.14 m$^2$), mostly because we wanted to minimize the intra-plot substrate variability, which is extremely important for bryophytes. Concerning bryophytes, such selection of relatively small sample plots agrees with the literature (Potter et al., 2013). However, we agree with the reviewer that using small plots can increase the probability of missing some (especially rare) species and potentially also increase the role of stochastic processes. Motivated by this reviewer comment, we repeated the analyses also with the bryophyte community sampled on the larger (100 m$^2$) plots (see results presented in the table below). The main conclusions of our study are fully supported by these new results, which basically mirror patterns found with the smaller plots. Interestingly, these new results further support our shift from the maximum VPD to mean VPD as the main explanatory variable. To conclude, we still prefer to base our results on smaller plots within the paper, mostly because of the possible issues with the substrate heterogeneity discussed above. However, we are ready to add the result based on larger plots either to the supplementary material or to the main text if the Editor prefer to do so.

| | | Species composition (db-RDA) | | Species richness (GAM) |
|---|---|---|---|---|
| | | Sørensen | Simpson | Number of species |
| **smaller plots** | **VPD max** | 10.95 % ** | 13.52 % ** | 31.2 % *** |
| | **VPD mean** | 16.09 % *** | 17.15 % *** | 32.8 % *** |
| **larger plots** | **VPD max** | 11.87 % ** | 10.21 % ** | 14.4 % *** |
| | **VPD mean** | 22.14 % *** | 13.10 % *** | 49.0 % *** |

*Table of results of db-RDA and GAM models – comparison for smaller and larger research plots*

**VPD-variability.**

I had a bit difficulty in flowing the text of how you calculated VPD-variability and when you talked about the variability over time and over space. And then what you take an average of. I think you need to carefully revise so that a reader understands all of this. For example how can you have a mean value of the standard deviation of the maximum value? And then you talk about range of plot means in Table 1. I am sure you have done it correctly it is just that it become difficult to follow when you have mean and SD values in a day, between days, between plots etc. Especially rows 122-124 I couldn't follow entirely, but revise also in other parts of the text and figure legends so that it is crystal clear when VPD variability consider spatial or temporal aspects for example.

Thank you for bringing this reader's perspective to our attention. We will do our best to make this clear in the text.

Within the paper, we mostly write about spatial VPD variability, which is the most important for the results presented in our paper. However, we agree that the different statistics of the VPD variability presented within the paper can be confusing for the reader. We presented these different statistics in order to describe the different aspects of the data used in our study. In the revision, we will thoroughly revise the text for clarity, and we will focus on the most important aspects of the data variability crucial for our results.

For clarification, we have only presented the variability over time on lines 157-160. On line 158, we talked about range and overall mean of raw measurements performed every 30 minutes. Table 1 reports ranges and means of microclimatic variables (the average daily maxima) used as explanatory variables in our multivariate analyses. We hoped that together, this information should help the readers to create a better picture of the data collected.

Regarding the spatial VPD variability. We express this spatial variability as the standard deviation (SD) of the plot-specific values. The mean value of these SD was calculated in two steps – first we calculated SDs for each individual day within the study period from daily maxima/mean measured at all study plots and then we averaged these daily SDs values over the whole study period. We already described this process on lines 122-124, but we will further revise the text in order to improve the clarity for the reader.

**Grouping of the bryophytes.**

Species could be grouped in many ways and you have three columns in Tablc C1: taxonomy, Major biome and Eastern limit. It seems in the results that you would like to say something on what is characterizing those that are sensitive to high VPD. However, in the results you have not really analysed the results in such a way and you instead talk about "small liverworts", "hygrophilous bryophytes", "suboceanic" and "mesic" species. And in several other places you talk about "azonal" species, which is a term not many readers will understand. Yet in other places you say "regionally rare species". You have so many terms and none of these categories are in Table C1. And you use words such as "in contrast" but these groups are not contrasts to each other in most cases but just different ways of describing them. The number of species you have is not very large so perhaps it can be difficult to divide them into several group for the analysis and it might be just enough to tell the general statistics on the community which you have done and present the results at the species level as in Figure 5. Then if you want you can exemplify species which are less and more sensitive, but perhaps don't need to put them into a category. Or select one or two "traits" and do a formal test. Substrate is another category that is often useful for describing bryophyte communities.

Thank you for this insight. You are right that it is difficult to put the studied bryophytes into several clearly defined categories for the formal analyses. We indeed wanted to highlight that the species most sensitive to VPD are the species whose occurrence in the studied area can be seen as unexpected with regard to the regional macroclimate (for such species occurrences we used the term azonal). These species are often species which are typical for more oceanic climates or species which mostly occur in the central European mountains. Joint occurrence of these species in exceptionally low elevation within the studied area always puzzled central European bryologists and nature conservationists. Here we found that i) these species are sensitive to atmospheric VPD, ii) occur predominantly in the sites with low VPD, and iii) therefore low VPD sites serve as their microclimatic refugia within otherwise unsuitable landscape matrix. In the revised paper, we will try to explain these important results more clearly, and we will reduce the number of categories used to refer to these species. Following your suggestions, we will focus more on individual species rather than on somewhat arbitrary defined species groups. The categories provided in Table C1 were meant as an attempt to summarize the distributional ranges of the species studied, but you are right that it was rather confusing since we have not used these categories anywhere else. As you

mentioned, the number of species we have is not very large, so it is difficult to strictly divide them into several groups for analysis. Therefore, we prefer to follow your suggestion and concentrate on the general statistics of the community and present and discuss the results at the species level (Figure 5). We will further unify the terms in the text, and we will discuss the results more on species - rather than group - level.

**Detailjed comments:**

Sensitivity of bryophytes to high maximum temperatures and high VPD. I think there are more references on this even if they might be more implicit. But for example various studies on forest edge effects on bryophytes could be relevant. Check also Dahlberg et al. 2020 in Environmental and Experimental Botany, who saw some interesting correlations with maximum temperature and distributions. Perhaps you might also be interested in Merinero et al. 2020 in Ecology who used evaporometers to capture the importance of VPD as a driver of bryophtye performans.

Thank you for your literature recommendation. We searched the literature thoroughly, but the studies exploring directly the effects of atmospheric VPD on forest bryophytes are surprisingly rare. Often, the link is indirect and supported by the measurement of the different variables, as in both references you suggested. Nevertheless, we re-read these references and agree that they are relevant for our study, therefore, we will refer to them in the revision.

Figure B1. Would it be good to indicate the 1:1 line in this graph and discuss a bit more on why your line is deviating. But very interesting that you have such a strong relationship!

Thank you for this suggestion. We will include the 1:1 line in Fig. B1 and will discuss the reasons for the overestimation of local VPD (deviation from 1:1 line) with this method.

**References**

Potter, K. A., Woods, H. A., and Pincebourde, S.: Microclimatic challenges in global change biology., Glob. Change Biol., 19(10), 2932–2939, https://doi.org/10.1111/gcb.12257, 2013.

---

## Author Response (AR1)

**Author response**

- We sincerely thank both reviewers for taking the time to thoroughly review our manuscript and for their positive comments and insightful suggestions. We also thank the editor for her careful consideration of our manuscript and helpfulness and support during the submission and review process.
- For clarity, reviewer's comments are in standard black font, author responses are in blue font and links to the main changes to the manuscript are in red font. The numbers of lines used in reviewer's comments refer to the originally submitted manuscript, the line numbers we refer to in this response correspond to the marked-up manuscript version.

**Response RC1: Alain Vanderpoorten**

I found this paper most original and useful as it addresses the very timely question of microclimates on diversity patterns, with a special emphasis on one key parameter: VPD. To give the paper the impact that it deserves and emphasize the relevance of VPD as an important, ecologically meaningful variable, I have two suggestions to make: (i) define VPD and explain, in the Introduction, what its ecological relevance as compared to other microclimatic variables such as T or RH alone (in other words, why would it be important to integrate VPD in ecological studies) and (ii) show that VPD is indeed a better predictor of species richness and composition than T or RH alone. This could be easily done by re-running the analyses, using T and RH as predictors, and showing that the use of VPD results in a higher percent variance of bryophyte richness and composition community explained.

Thank you for the positive assessment of our manuscript and useful comments and suggestions. Regarding your first suggestion (i), we added a definition of VPD to the Introduction, and we added an explanation of its ecological relevance compared to temperature and relative humidity. We also emphasized and explained in more detail the reasons why it is important to integrate VPD into ecological studies instead of, e.g. the temperature itself or relative humidity.

• VPD definition: L39-40, L62-63

- explanation of VPD ecological relevance compared to T and RH: L40-47
- main reasons for integrating VPD into ecological studies are discussed throughout the Introduction (e.g. L48-62, L70-74, L78-83, L86-91)

Regarding the second suggestion (ii), these three variables (VPD, T, RH) are closely corelated in our study area (see correlation matrix on page 2 in our previous response AR1 <a href="https://doi.org/10.5194/egusphere-2025-1244-AC1">https://doi.org/10.5194/egusphere-2025-1244-AC1</a> or Appendix A, Fig. A1 in the revision). Therefore, we cannot statistically fully separate their effects, which we clearly stated in the revision (L212). It is indeed difficult to separate the effects of these variables with observation data, and we expanded the discussion of this topic in the revision (L357-372, L455-480).

However, based on a careful study of literature (e. g. Eamus et al., 2013; Grossiord et al., 2020; Fu et al., 2022; Novick et al., 2024), we agree with these authors that VPD is the most physiologically relevant variable (as we already discussed on lines 239-255 and 270-287 in the original manuscript and now also in more depth in the revision, e. g. L40-41, L46-47, L78-79 and elsewhere throughout the Introduction and in section 4.2 in the Discussion). VPD directly expresses the driving force of evaporation, because it integrates both temperature and air humidity in a theoretically and physically preferable way (Anderson 1936). In contrast, relative humidity alone indicates very different atmospheric moisture conditions at different temperatures. An atmosphere with the same RH may be very "dry" (when the temperature is high) or it may be very "wet" (when the temperature is low). Thus, RH alone does not indicate the atmospheric moisture conditions in a physiologically meaningful

way. As Campbell and Norman (1997) stated in their famous book "An Introduction to Environmental Biophysics" (page 49): "It is perhaps unfortunate that one of the most common measurements of atmospheric moisture is relative humidity. The measurement itself is essentially useless as an environmental variable except as a means, along with air temperature, of obtaining the vapor pressure, mole fraction, or dew point temperature." We fully agree with this view and therefore we decided not to use RH as a separate environmental variable in the revision. We summarize the reasons for this decision on L40-47.

In the case of maximum air temperature, our previous studies showed that maximum air temperature is the most relevant factor linked to evaporative stress both for forest vascular plants (Macek et al., 2019) and forest bryophyte communities (Man et al., 2022). Unfortunately, none of these studies measured VPD. Therefore, in the revision we focused not only on the VPD but also on the maximum temperature, because all these microclimatic variables represent evaporative stress. Nevertheless, atmospheric VPD consistently explained more variability in species composition than either T or RH (as we showed in table on page 3 in our previous response AR1 https://doi.org/10.5194/egusphere-2025-1244-AC1).

For reasons given above, in the revision we also directly analyzed effects of the maximum air temperature and clearly indicated that the VPD is closely related to the temperature within our study region (L340-346, Appendix A). Nevertheless, while thinking about this reviewer comment, we realized that we should also explore not only the effects of maximum VPD but also mean VPD. We initially decided to use the maximum VPD, because maximum temperature, which practically represented maximum VPD in our study area (Pearson correlation between VPDmax and Tmax = 0.98), was the most important microclimatic variable affecting species composition of vascular plants (Macek et al., 2019) as well as bryophytes (Man et al., 2022) in the central European region. However, thanks to the reviewer's second comment (ii), we have renewed the discussion in our author team, and we have decided to reanalyze our data also with the mean VPD. The main motivation for this change was that the averages better capture the long-term characteristics of site microclimatic conditions, which can be more important for bryophytes than short-term extremes captured by the maximum VPD (which can however be more relevant for vascular plants as they lack bryophyte's unique ability to tolerate desiccation). Interestingly, using the mean VPD instead of maximum VPD resulted in substantial increase in the explained variation, both in bryophyte community composition as well as species richness (Table 2).

Therefore, we decided to analyze both mean and maximum VPD in the revision. This change allows us to show the reader our thought process in selecting microclimatic variables representing evaporative stress, to explore the role of mean and maximum VPD and maximum temperature, and to better support our claims. At the same time, this change did not affect the conclusions about the importance of temperature as the main driver of VPD variability across the landscape.

- we added correlation matrix to Appendix A to show the correlation between microclimatic variables representing evaporative stress
- we explored the relationship of the main gradients in species composition and richness with variables representing evaporative stress (VPDmean, VPDmax and Tmax), see L195-200 and Fig. 2
- we directly tested the effects of variables representing evaporative stress (VPDmean, VPDmax and Tmax) on species composition and richness, see L201-206, L207-211 and Tab. 2
- to disentangle the effects of atmospheric VPD from the effects of the maximum temperature, we partitioned the variation in species composition and richness explained by mean VPD and maximum temperature into independent and shared fractions, L212-226 and Fig. 3
- we expanded the discussion about the correlated effects of microclimatic variables representing evaporative stress in the Discussion, see L357-372 and section 4.3

One of the first results being discussed is the tight relationship between VPD and Tmax-it does not seem to me that this results directly stems out the analyses presented? Could this relationship be actually evidenced based on the data generated?

Yes, based on our data, we have evidenced a close relationship between VPD and T. This result stems from the analyses presented (Results part 3.1 VPD variability, second paragraph in the original manuscript). However, it is true that we do not directly report the relationship between VPD and T, but instead the relationship between VPD and saturation vapor pressure (Psat), which is solely a function of temperature (as we stated on line 43 and 109 in the original manuscript), so the relationship between VPD and Psat also illustrates the relationship between VPD and T. In the revision, we further highlighted this relationship in order to make it more understandable for the reader. However, we decided not to add to the Appendix a plot directly showing the relationship between VPD and T (see on page 3 in our previous response AR1 <a href="https://doi.org/10.5194/egusphere-2025-1244-AC1">https://doi.org/10.5194/egusphere-2025-1244-AC1</a>). Instead, we revised Figure 3 (in the revision Figure 6) and Figure 2 which now shows the relationships between maximum VPD, mean VPD and maximum T, added correlation matrix showing the correlation between VPD and Tmax to Appendix A and added results of variation partitioning based on a multiple linear regression model with mean VPD as the response variable and mean Psat and Pair as the predictors.

- see L64-65
  - see L258-261 and Fig. 6
- see L391

In the meantime, if VPD is tightly correlated with Tmax, does this not slightly undermine the premises of the study, that is, the potential benefit of an integrative variable such as VPD in ecological studies as compared to a 'simple' variable like Tmax?

While we indeed found that VPD is driven by saturated vapor pressure (respectively T), and the contribution of actual vapor pressure is only small, the situation at larger spatial scales or near water bodies may be different (discussed on L379-382). Under these conditions, the role of actual vapor pressure may be more prominent, and T alone may no longer be a good proxy of VPD. Therefore, we cannot recommend the use of a "simple" variable such as T instead of VPD. Generally, VPD is more physiologically relevant than T (as we discuss within the original manuscript) and even if we found that the T is closely correlated to VPD in our study area, this correlation can change under different conditions elsewhere. Therefore, we would still recommend measuring VPD directly, but also acknowledge that on landscape and finer scales, gradients in VPD and temperature can be very closely correlated and therefore difficult to distinguish. Nevertheless, our new results of variation partitioning of species composition and richness showed that maximum temperature does not have any unique effect independent of VPD effect and that mean atmospheric VPD explains significantly more variation in species composition and richness than maximum temperature.

• see L212-226, Fig. 3 and L366-L39

Since the second main result discussed is that it is possible to estimate VPD from local T measurements with HR measured as nearby weather stations, I suggest moving the content of this appendix into the result section of the main text. Would that mean that one can efficiently characterize microclimates using temperature sensors only, which are much cheaper than sensors combining T+HR?

We considered including the estimation of VPD from local T measurements with air humidity measured at a nearby weather station in the main results, but we decided not to do it in the revision because this is more of an application of our results rather than a hypothesis we have had at the beginning of our research. Moreover, it would even more expand the Methods. However, we are prepared to move these results into the main text, if the editor prefers to do so.

The answer to the second part of your question, whether that would mean that one can efficiently characterize microclimates using temperature sensors, is partly discussed in our response to your previous question about using simple variables like T instead of VPD. Our approach to the VPD estimation can indeed be useful in cases where air humidity measurements are not available, but this should be done carefully and should be more

widely tested across different vegetation types and spatial scales. Moreover, as we stated in the Discussion
(L379-382) and in Appendix B, this approach does not provide a reliable estimate of absolute local atmospheric
VPD, but rather the relative position of the locality on the VPD gradient. So, we would not recommend
characterizing microclimate using temperature sensors only.

I was a bit surprised by the relatively limited contribution of VPD (about 11%) to variation in species composition among plots, whereas the introduction rightly emphasizes that in poikilohydric organisms like bryophytes, one could expect VPD to be a prime factor driving community composition. Looking at Fig1, one would think that communities at the top of a cliff would be very different from those in buffered conditions. I wonder whether this could be due to the fact that, as Fig1 suggests, this is a very rugged terrain, and that there is hence a huge (intra-plot) micro-habitat variation that is actually more important than (inter-plot) microclimatic variation. More information on the sampling sites would be welcome to understand the spatial

heterogeneity potentially present in the sampling plots.

Compositional variation explained by a multivariate analysis depends on dataset internal heterogeneity and therefore it is not comparable among datasets (Økland, 1999). Especially in ecological studies covering different vegetation types with limited overlap of the species composition between plots (as in our case), the percentage of the compositional variation explained by one variable is typically in similar range (Økland, 1999). To put the percentage of the explained variation further into perspective, the best theoretically possible variable can explain max. of 30 % of compositional variation in our dataset (for methodological explanation see e.g. Macek et al. 2019). We decided not to complicate the paper with the reporting of this maximum explainable compositional variability which will require additional description of the methods used.

Moreover, motivated by the reviewer comment about the relevance of maximum VPD, we recalculated our results also with mean VPD, which resulted in the substantial increase in the compositional variation explained by our analyses (Table 2). We are therefore confident that these results support our conclusion that atmospheric VPD is an important driver of bryophyte species composition across the landscape.

It would help the reader if it was reminded in the Result section based on which analysis each result was obtained. For example, the variation partitioning and db-RDA are not mentioned in the Result section, and mentioning them would help the reader making a link between the M&Ms and results. For example, I am not sure which analysis was implemented to reach the result described L179-180 ('ecological relevance of VPD as compared to HR alone').

In the results, we clearly stated which results are based on which analysis (e. g. L272-273, L303-305, L314-316). Specifically, the results presented on lines 179-180 were obtained by the *envfit* function from the *vegan* R package with 999 random permutations.

In the discussion, it would be interesting to add a section explaining what could be the factors accounting for the spatial variation of VPD reported, and why Pair exhibits such a comparatively narrow range of variation. At present the discussion falls a bit short—especially since the entire §4.2 (from L239 onwards) actually belongs to the Introduction (why bryophytes would be sensitive indicators of VPD variation), and not to the Discussion as it does not help interpreting the results presented.

Thank you for the suggestion. We tried to move the whole section 4.2 from the Discussion to the Introduction, but we think it is too detailed for the general Introduction, so we moved it only partly (L78-79, L82-83). However, we think the information from section 4.2 are interesting and serve as a good context for the Discussion, especially when we connected this information from literature to our results of reported VPD condition in our study area (L433-435). Therefore, we prefer to leave most of this section in the Discussion. Regarding the factors accounting for the spatial variation of VPD, we already discussed them in section 4.1 of the Discussion (lines 223-230), but we substantially expanded this discussion in the revision (L391-395, L399-407).

| 179                             | Cited references                                                                                                                                                                                                                                                                                                                                                                                                                                                                                              |
|---------------------------------|---------------------------------------------------------------------------------------------------------------------------------------------------------------------------------------------------------------------------------------------------------------------------------------------------------------------------------------------------------------------------------------------------------------------------------------------------------------------------------------------------------------|
| 180
181                      | Anderson, D. B.: Relative humidity or vapor pressure deficit, Ecology, 17(2), 277–282, <a href="https://doi.org/10.2307/1931468">https://doi.org/10.2307/1931468</a> , 1936.                                                                                                                                                                                                                                                                                                                                  |
| 182
183
184               | Eamus, D., Boulain, N., Cleverly, J., and Breshears, D. D.: Global change-type drought-induced tree mortality: vapor pressure deficit is more important than temperature per se in causing decline in tree health, Ecol. Evol., 3(8), 2711–2729, <a href="https://doi.org/10.1002/ece3.664">https://doi.org/10.1002/ece3.664</a> , 2013.                                                                                                                                                                      |
| 185
186
187               | Fu, Z., Ciais, P., Prentice, I. C., Gentine, P., Makowski, D., Bastos, A., Luo, X., Green, J. K., Stoy, P. C., Yang, H., and Hajima, T.: Atmospheric dryness reduces photosynthesis along a large range of soil water deficits, Nat. Commun., 13(1), 1–10, <a href="https://doi.org/10.1038/s41467-022-28652-7">https://doi.org/10.1038/s41467-022-28652-7</a> , 2022.                                                                                                                                        |
| 188
189                      | Grossiord, C., Buckley, T. N., Cernusak, L. A., Novick, K. A., Poulter, B., Siegwolf, R. T. W., Sperry, J. S., and McDowell, N. G.: Plant responses to rising vapor pressure deficit, New Phytol., 226(6), 1550–1566, <a href="https://doi.org/10.1111/nph.16485">https://doi.org/10.1111/nph.16485</a> , 2020.                                                                                                                                                                                               |
| 190
191                      | Macek, M., Kopecký, M., and Wild, J.: Maximum air temperature controlled by landscape topography affects plant species composition in temperate forests, Landscape Ecol., 34, 2541–2556, <a href="https://doi.org/10.1007/s10980-019-00903-x">https://doi.org/10.1007/s10980-019-00903-x</a> , 2019.                                                                                                                                                                                                          |
| 192
193
194               | Man, M., Wild, J., Macek, M., and Kopecký, M.: Can high-resolution topography and forest canopy structure substitute microclimate measurements? Bryophytes say no., Sci. Total Environ., 821, 153377, <a href="https://doi.org/10.1016/j.scitotenv.2022.153377">https://doi.org/10.1016/j.scitotenv.2022.153377</a> , 2022.                                                                                                                                                                                   |
| 195
196
197
198        | Novick, K. A., Ficklin, D. L., Grossiord, C., Konings, A. G., Martínez-Vilalta, J., Sadok, W., Trugman, A. T., Williams, A. P., Wright, A. J., Abatzoglou, J. T., Dannenberg, M. P., Gentine, P., Guan, K., Johnston, M. R., Lowman, L. E. L., Moore, D. J. P., and McDowell, N. G.: The impacts of rising vapour pressure deficit in natural and managed ecosystems, Plant Cell Environ., 47(9), 3561-3589, <a href="https://doi.org/10.1111/pce.14846">https://doi.org/10.1111/pce.14846</a> , 2024.        |
| 199
200                      | Økland, R.H.: On the variation explained by ordination and constrained ordination axes, J. Veg. Sci., 10, 131–136, <a href="https://doi.org/10.2307/3237168">https://doi.org/10.2307/3237168</a> , 1999.                                                                                                                                                                                                                                                                                                      |
| 201                             |                                                                                                                                                                                                                                                                                                                                                                                                                                                                                                               |
| 202                             | Response RC2: Anonymous                                                                                                                                                                                                                                                                                                                                                                                                                                                                                       |
| 203                             | Dear authors,                                                                                                                                                                                                                                                                                                                                                                                                                                                                                                 |
| 204
205
206
207
208 | I very much enjoyed reading your manuscript and think you have done a good job in both setting the question, conducting the research and writing the paper. It was easy to read and I think you have many strong points to raise including the strong effects of VPD, that it is the saturation that varies, the strong link to maximum temperature and what that implies for our possibility to monitor and study the effects of VPD. Moreover you have illustrated that well with a data set of bryophytes. |
| 209
210                      | I will try to be constructive to point out a few things that perhaps could improve the clarity of the paper. In general I think you could take a careful look at the flow of the text to avoid repetition and increase clarity.                                                                                                                                                                                                                                                                               |
| 211
212                      | We thank the reviewer for such a positive evaluation and useful feedback on our manuscript. We did our best to incorporate the suggestions and improve the clarity and fluency of the text.                                                                                                                                                                                                                                                                                                                   |
| 213
214
215               | At the same time, we would like to draw your attention to a changes we did based on suggestions from reviewer RC1 – most importantly, we used also mean VPD together with maximum VPD and maximum temperature in the revision.                                                                                                                                                                                                                                                                                |
| 216                             | Design of the study.                                                                                                                                                                                                                                                                                                                                                                                                                                                                                          |
| 217
218
219
220        | There is always a trade-off related to the size of the plot to study. A small plot as yours is good to capture the microclimate at one spot, but you will miss a lot of rare species in the landscape. In your case you have a circular plot of 1 meter radius, right? If wo it could be good to spell out. I lack some information on how you selected sites. Was it done using maps and satellite images and getting a coordinate from there? How did you                                                   |

select them in the field? What if a tree was in the plot? Or a big boulder? Did you make any notes on substrate? Substrate composition is often an important driver of species composition of bryophytes. You have an ambitious approach of covering the whole forest landscapes and then perhaps your sample size of 38 plots is a bit low. But you got very interesting results and have an interesting approach so I am still fine with this.

Yes, the species data presented in our manuscript were recorded on a circular plot of 1 meter radius. These plots were selected through stratified-random sampling to capture the main microclimatic gradients within the core zone of the national park. Specifically, using GIS and detailed digital terrain model, we first divided the core zone into geographical strata defined by the position on the terrain (valley bottoms, lower slopes, upper slopes and ridges) and further separated the slopes into the slopes with predominantly northern and southern orientation. Then, we used GIS algorithms to randomly sample the equal number of locations within each defined strata with the additional conditions that the sampled locations must be separated by at least 50 m.

In the field, we navigated to the selected location with GPS device and placed the center of the plot 1.5 m to the north from the nearest tree. This tree was later equipped with the HOBO datalogger for air temperature and humidity measurements. Additional condition for plot selection in the field was that the circular area with a 1 m radius around the plot center must not contain any rocks or big stones in order to reduce the within plot substrate heterogeneity.

We fully agree that it would be nice to have more plots with complete data (both in situ measured microclimate and sampled bryophytes). However, we think that the 38 plots used in this study is sufficient for our aims. As we described above, the plots were carefully selected through stratified-random sampling. Therefore, they provide representative sample of the environmental variability within the core zone of the national park. The potential effects of within plot substrate heterogeneity were further reduced by the additional criteria for the plot selection (specified above).

Regarding the size of the research plots - we discussed it a lot, because we also collected larger (100 m2) plots in each measurement site. The smaller (3.14 m2) plots were always nested in the center of the larger 100 m2 plot. However, during the bryophyte sampling at the larger plot, we did not make detailed records of the substrate, so we finally decided to based our analyses on the smaller plots (3.14 m2), mostly because we wanted to minimize the intra-plot substrate variability, which is extremely important for bryophytes. Concerning bryophytes, such selection of relatively small sample plots agrees with the literature (Potter et al., 2013). However, we agree with the reviewer that using small plots can increase the probability of missing some (especially rare) species and potentially also increase the role of stochastic processes. Motivated by this reviewer comment, we repeated the analyses also with the bryophyte community sampled on the larger (100 m²) plots (see results presented in the table on the page 2 in our previous response AR2 https://doi.org/10.5194/egusphere-2025-1244-AC2). The main conclusions of our study are fully supported by these new results, which basically mirror patterns found with the smaller plots. Interestingly, these new results further support our shift from the maximum VPD to mean VPD as the main explanatory variable. To conclude, we still prefer to base our results on smaller plots within the paper, mostly because of the possible issues with the substrate heterogeneity discussed above. However, we are ready to add the result based on larger plots either to the supplementary material or to the main text if the Editor prefers to do so.

- information about the selection of the research plots: L 129-134
- specification of the plot size: L142

**VPD-variability.**

I had a bit difficulty in flowing the text of how you calculated VPD-variability and when you talked about the variability over time and over space. And then what you take an average of. I think you need to carefully revise so that a reader understands all of this. For example how can you have a mean value of the standard deviation of the maximum value? And then you talk about range of plot means in Table 1. I am sure you have done it

correctly it is just that it become difficult to follow when you have mean and SD values in a day, between days, between plots etc. Especially rows 122-124 I couldn't follow entirely, but revise also in other parts of the text and figure legends so that it is crystal clear when VPD variability consider spatial or temporal aspects for example.

Thank you for bringing this reader's perspective to our attention, we tried to do our best to improve the description of our approach and results for VPD variability in the revised text. In the revision, we thoroughly revised the text for clarity, and we focused only on the most important aspects of the data variability crucial for our results.

To avoid readers' confusion, we did not present the variability over time in the revision, as these aspect was not necessary for the understanding of our results and served more as a meta-information for the reader (in the original manuscript on lines 157-160). We also revised Table 1 in order to present an overview of the microclimatic variables used in the analyses. These changes will hopefully provide a more structured and more easily understandable overview of our data and approach we used to analyze them.

Regarding the spatial VPD variability: We express this spatial variability as the standard deviation (SD) of plot-specific values. The mean value of these SD was calculated in two steps – first we calculated SDs for each individual day within the study period from daily values measured at all study plots and then we averaged these daily SDs values over the whole study period. We described this process on lines 122-124 in the original manuscript, but we revised the text to improve the clarity in the revision.

- see Table 1 (overview of microclimatic variables representing evaporative stress and their components)
- see updated section 2.4.2 (L247-257)
- see updated Fig. 5 and Fig. 6
- we removed part about the VPD variability over time from the main text (L332-334), and therefore we also removed original Fig.A1 and Fig. A2 from the Appendix A

**Grouping of the bryophytes.**

Species could be grouped in many ways and you have three columns in Table C1: taxonomy, Major biome and Eastern limit. It seems in the results that you would like to say something on what is characterizing those that are sensitive to high VPD. However, in the results you have not really analysed the results in such a way and you instead talk about "small liverworts", "hygrophilous bryophytes", "suboceanic" and "mesic" species. And in several other places you talk about "azonal" species, which is a term not many readers will understand. Yet in other places you say "regionally rare species". You have so many terms and none of these categories are in Table C1. And you use words such as "in contrast" but these groups are not contrasts to each other in most cases but just different ways of describing them. The number of species you have is not very large so perhaps it can be difficult to divide them into several group for the analysis and it might be just enough to tell the general statistics on the community which you have done and present the results at the species level as in Figure 5. Then if you want you can exemplify species which are less and more sensitive, but perhaps don't need to put them into a category. Or select one or two "traits" and do a formal test. Substrate is another category that is often useful for describing bryophyte communities.

You are right that it is difficult to put the studied bryophytes into several clearly defined categories for the formal analyses. We indeed wanted to highlight that the species most sensitive to VPD are the species whose occurrence in the studied area can be seen as unexpected with regard to the regional macroclimate (for such species occurrences we used the term azonal in the original submission, in the revision we refer to these species as "species with disjunct occurrence in central Europe" or "species near their distributional range limits", which is hopefully more understandable). These species are often typical for more oceanic climates or species which mostly occur in the central European mountains. Joint occurrence of these species in exceptionally low elevation within the studied area always puzzled central European bryologists and nature

conservationists. Here we found that i) these species are sensitive to atmospheric VPD, ii) occur predominantly in the sites with low VPD, and iii) therefore low VPD sites serve as their microclimatic refugia within otherwise unsuitable landscape matrix.

In the revision, we did our best to explain these important results more clearly. Further, we tried to reduce the number of categories used to refer to these species and we removed the biogeographical categories from Table C1 to prevent the confusion. Following your suggestions, we focused more on individual species rather than on somewhat arbitrary defined species groups (L317-321). As you mentioned, the number of species we have is not very large, so it is difficult to strictly divide them into several groups for formal analysis. Instead, we formulated two hypotheses about the community structure and co-occurrence patterns along the VPD gradient and tested these hypotheses with the nestedness analyses in the revised manuscript. First, we tested the hypothesis that the bryophyte communities from sites with high VPD are nested subsets of bryophyte communities from sites with low VPD. Second, we tested the hypothesis that more frequent bryophyte species occur along the whole VPD gradient, but less frequent species are concentrated on sites with low VPD. Our new results supported both tested hypotheses and provide additional evidence about the importance of atmospheric VPD for bryophyte community structure.

- see e. g. L27, L35, L495
- nestedness analysis: Methods L187-188, L227-242, Results L312-316
- see updated Fig. 4

**Detailed comments:**

- Sensitivity of bryophytes to high maximum temperatures and high VPD. I think there are more references on this even if they might be more implicit. But for example various studies on forest edge effects on bryophytes could be relevant. Check also Dahlberg et al. 2020 in Environmental and Experimental Botany, who saw some interesting correlations with maximum temperature and distributions. Perhaps you might also be interested in Merinero et al. 2020 in Ecology who used evaporometers to capture the importance of VPD as a driver of bryophtye performans.
- Thank you for your literature recommendation. We searched the literature thoroughly, but the studies exploring directly the effects of atmospheric VPD on forest bryophytes are surprisingly rare. Often, the link is indirect and supported by the measurement of the different variables, as in both references you suggested. Nevertheless, we re-read these references and agree that they are relevant for our study, therefore, we referred to them in the revision (L376 and L424).
- Figure B1. Would it be good to indicate the 1:1 line in this graph and discuss a bit more on why your line is deviating. But very interesting that you have such a strong relationship!
- Thank you for the suggestion. We included the 1:1 line in Fig. B1 and discussed more the reasons for the general overestimation of local VPD (positive deviation from 1:1 line), both in the Appendix B and in the Discussion (L379-382).

**346 References**

Potter, K. A., Woods, H. A., and Pincebourde, S.: Microclimatic challenges in global change biology., Glob. Change Biol., 19(10), 2932–2939, <a href="https://doi.org/10.1111/gcb.12257">https://doi.org/10.1111/gcb.12257</a>, 2013.